# Progressive Online Video Understanding with Evidence-Aligned Timing and Transparent Decisions

**Kecheng Zhang[1] Zongxin Yang[2] Mingfei Han[1,3] Haihong Hao[1] Yunzhi Zhuge[4] Changlin Li[5]**
**Junhan Zhao[6] Zhihui Li[1][\*] Xiaojun Chang[1]**

[1] University of Science and Technology of China [2] Harvard University [3] Department of CV, MBZUAI
[4] Dalian University of Technology [5] Stanford University [6] Chicago University

## Abstract

Visual agents operating in the wild must respond to queries precisely when sufficient evidence first appears in a video stream, a critical capability that is overlooked by conventional video LLMs evaluated in offline settings. The shift to an online, streaming paradigm introduces significant challenges: a lack of decision transparency, the difficulty of aligning response timing with visual evidence, and the need to maintain a global, causally consistent understanding under tight computational budgets. To address these issues, we propose a novel framework that decouples reasoning control from memory integration. We introduce **Thinking-QwenVL**, an instantiation of this framework with two core components. First, the *Active Thinking Decision Maker (ATDM)* is a transparent reasoning controller that externalizes its decision process using observable progress ($\rho$) and confidence ($c$) metrics. This allows it to precisely time its response $t_r$ to match the first-sufficient-evidence timestamp $t^\star$ while streaming its reasoning to the user. Second, the *Hierarchical Progressive Semantic Integration (HPSI)* module acts as an efficient memory system. It employs a set of learnable, multi-level aggregation tokens that are propagated across clips to build a rich, global cognitive state without exceeding token budgets. Extensive experiments demonstrate the effectiveness of ATDM and HPSI, e.g., Thinking-QwenVL improves the accuracy of the previous state-of-the-art from 67.63% to 71.60% on the StreamingBench benchmark.

## 1 Introduction

Visual evidence-aligned response timing is central to visual agents operating in the wild: an assistant should answer only once the video first contains sufficient evidence, and it should show *when* and *why* (Cai et al., 2025; Subramanian et al., 2024). Consider a domestic robot asked, "is the kettle boiling?" It should *wait for* visible steam or a rolling boil and report immediately *at the first frame these signals appear* to avoid danger (Li et al., 2019). A driver-assistance agent queried, "is it safe to turn right?" must defer until the crosswalk and signal are jointly favorable.

Despite rapid progress, representative video-understanding LLMs such as VideoLLaMA3 (Zhang et al., 2025), InternVL3 (Zhu et al., 2025), and Qwen2-VL (Wang et al., 2024a) are commonly evaluated in idealized *offline* regimes. The entire video is preloaded; frames or clips may be retrieved and re-encoded multiple times; and global reasoning precedes response generation. This practice diverges from interactive, real-world operation in which users ask at time $t_q$, but the earliest sufficient evidence may not appear until $t^\star$. A system should respond at $t_r$ only when $t_r \approx t^\star$; otherwise, avoidable compute and queuing delays degrade responsiveness and user experience. These issues motivate the *online video understanding* setting, which constrains the model to act only on currently accessible visual evidence while enabling perceivable and controllable interaction.

In online use, three aspects become decisive. *First, decision transparency and real-time feedback.* Collapsing timing into a black-box gate ("answer" vs. "defer") leaves no visibility into timestamps,

---
[\*]Corresponding authors.

intermediate conclusions, or progress, undermining controllability and trust during streaming interaction. *Second, evidence-aligned response timing.* With $t_q$, $t_r$, and $t^\star$ as defined above, the goal is to minimize $\delta = |t_r - t^\star|$ under streaming uncertainty and latency constraints without sacrificing correctness; recent benchmarks (e.g., OVOBench (Niu et al., 2025), RTVBench (Xun et al., 2025)) stratify tasks by the relation between $t_q$ and $t^\star$, yet many systems fix $t_r = t_q$ or use centered windows. *Third, global, causal updates under tight budgets.* Let $\mathbb{V}_t = \{v_1, \ldots, v_t\}$ denote the observed stream and $h_t$ a compact cognition state summarizing entities, events, and relations supported by $\mathbb{V}_t$. As new clips arrive, the model should revise hypotheses and propagate temporal/spatial constraints *globally*—not merely apply myopic, clip-local updates that break the storyline or causal consistency.

We address these needs with two complementary ideas that separate *reasoning control* from *memory/integration*. **i) Evidence-aligned, transparent timing (reasoning controller).** We replace a single opaque gate with a multi-stage, observable decision process that surfaces evidence-aligned timestamps, stage-wise progress $\rho$, concise rationales, and an estimated response time $t_r$; the controller self-triggers cross-clip reflection when confidence $c$ is low, so users can see *why now* or *why wait*. **ii) Progressive and global causal state (memory & integration)** with evolving visual evidence. We maintain and refine a compact, relation-aware $h_t$ under token/latency budgets so that cross-clip evidence updates the *global* understanding as the stream unfolds. The online framework proceeds stepwise: ingest the next clip and update $h_{t+1}$; the controller consults $(h_{t+1}, q)$, advances $\rho$ and $c$, and decides to answer (emitting $t_r$) or to wait/reflect; timestamps and interim conclusions are streamed to users for auditable, real-time interaction.

Building on these ideas, we present **Thinking-QwenVL**, which instantiates the framework with two modules. *Active Thinking Decision Maker* (ATDM) implements the controller: it factorizes timing into sub-goals with observable progress $\rho$ and confidence $c$, predicts an evidence-aligned $t_r$ via the quantitative indicators $(\rho, c)$, and self-triggers cross-clip reflection when needed. In doing so, it streams timestamps, interim conclusions, and rationale snippets to the user in real time, decision-making becoming *transparent*, *observable*, and *quantifiable*—with real-time progress and response feedback. *Hierarchical Progressive Semantic Integration* (HPSI) implements memory and integration inside the vision–language decoder: at multiple decoder depths (e.g., lower/middle/upper thirds), it inserts a small set of learnable multi-level aggregation tokens $p$ that attend to frame/clip tokens via structured sparse attention. The $p$ tokens are *carried forward* across clips as part of $h_t$ that is refined as new clips arrive, enabling causal, relation-preserving updates to the global visual view without inflating the token budget.

We evaluate on benchmarks designed for online video understanding, including Streaming-Bench (Lin et al., 2024), OVOBench (Niu et al., 2025), OVBench (Huang et al., 2024), and RTVBench (Xun et al., 2025), where Thinking-QwenVL attains strong results due to HPSI and ATDM of **71.6%**, **46.9%**, **35.6%**, and **35.9%**, respectively. Thinking-QwenVL also maintains competitive long-video performance—up to **67.7%** on VideoMME (Fu et al., 2024) and **68.3%** on MLVU (Zhou et al., 2024)—primarily due to HPSI that enables segment-wise attention perception and cross-clip causal relations preservation. In summary, our contributions are:

- We formalize evidence-aligned timing in the online regime via $(t_q, t_r, t^\star)$ and deviation $\delta$, elevate *decision transparency* to a first-class objective for streaming interaction, and propose a two-part framework Thinking-QwenVL for online video understanding.

- Combining ATDM and HPSI, we instantiate the framework with a controller that exposes $\rho$ and $c$ and aligns $t_r$ to first-sufficient evidence $t^\star$ with *self-triggered* reflection, and a hierarchical integration module with learnable multi-depth, multi-level aggregation tokens $p$ that guides segment-wise attention enhancement and preserves cross-clip relations, enabling globally consistent updates of $h_t$ under tight budgets.

## 2 RELATED WORK

**Offline Long Video Understanding.** Research on long-form video understanding investigates how to process vast numbers of visual tokens within limited context windows and constrained compute. Recent efforts have extended capability from short clips to videos exceeding ten minutes (Shen et al., 2024; Xue et al., 2024; Wang et al., 2024c; Zohar et al., 2024). Representative lines include adapting image-centric LMMs to long videos (e.g., LongVA building on LLaVA (Zhang et al., 2024b; Liu et al., 2023)), retrieval over graph/tree indices to shorten effective context (VideoRAG (Luo et al., 2024), Omni-AdaVideoRAG (Xue et al., 2025)), and improved temporal selection and training curricula (VideoLLaMA3 with differential frame pruning and vision-centric multi-stage train-

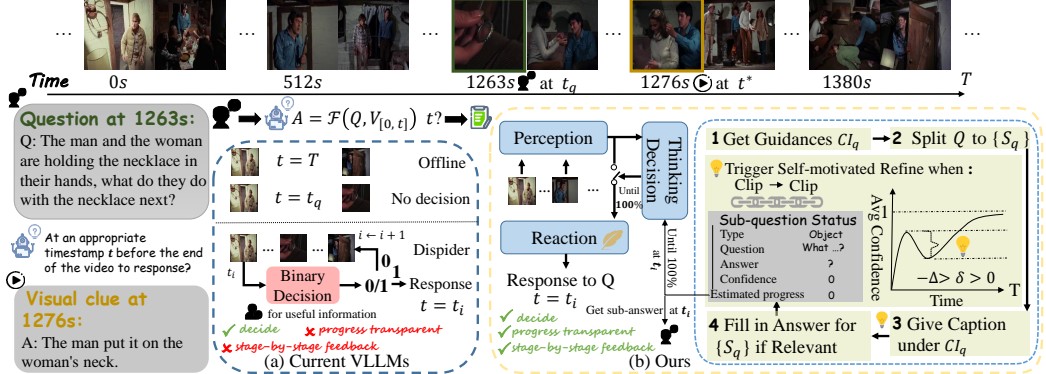

Figure 1: **Comparing paradigms vs. Ours.** Given a query $Q$, offline VLLMs answer only after the full video is available ($t = T$), while streaming models answer at the query moment ($t = t_q$); neither ensures *evidence-aligned* timing with the earliest evidence time $t^\star$. Our method decomposes $Q$ into sub-goals and maintains a progress estimate $\rho$, emitting real-time, stage-wise feedback at every step and selecting a response time $t_r \approx t^\star$, thereby reducing latency without sacrificing correctness and avoiding information-vacuum waiting.

ing (Zhang et al., 2025)). InternVL3 (Zhu et al., 2025) further explores *variable visual position encoding* and *text–time scaling* to better align temporal and textual streams. While recent advances improve offline reasoning over long videos, most methods assume full-video access and prioritize token reduction. So, offline pipelines sidestep interaction-critical needs: *evidence-aligned response timing*. These gaps motivate our online formulation, which preserves progressive understanding and couples inference with timely feedback.

**Online Video Understanding.** To better define and evaluate online video understanding, recent benchmarks such as OVOBench, StreamingBench, and RTVBench (Niu et al., 2025; Lin et al., 2024; Xun et al., 2025) have initiated systematic investigations in open-source settings. Existing methods largely split into two families. In fixed-response streaming (simply $t_r = t_q$), Stream-Bridge (Wang et al., 2025), StreamChat (Xiong et al., 2025), VideoStreaming (Qian et al., 2024), Flash-VStream (Zhang et al., 2024a), and VideoLLM-Online (Chen et al., 2024a) mainly optimize streaming readout, alignment, and memory, but do not make decision or align $t_r$ to $t^\star$. In timestamp-deciding methods, Dispider (Qian et al., 2025) compresses incoming clips and applies a binary head for answerability, yet the decision is opaque, repeatedly invoked without a principled stopping rule, and prone to prolonged non-answerable states that appear stalled to users; Timechat-Online (Yao et al., 2025) ties answerability to scene transitions, but scene change does not guarantee sufficient evidence, rendering it brittle and threshold-sensitive. By contrast, our formulation provides evidence-aligned timing and transparent decision progress, directly addressing these limitations. We employ the same single-pass, single-turn streaming regime rather than the multi-round video processing described in StreamBridge (Wang et al., 2025) to align with traditional streaming methods.

## 3 THINKING-QWENVL

**Overview.** We pursue *visual evidence-aligned, progressive, causal* understanding of a video stream. Let $\mathbb{V}_t = \{v_1, \ldots, v_t\}$ be the visible clips and $h_t$ a compact cognition state. With each new clip $v_{t+1}$, **HPSI** updates the state via $h_{t+1} = \mathcal{U}(h_t, v_{t+1})$, using a small set of multi-depth aggregation tokens with structured sparse attention to aggregate locally, integrate hierarchically, and propagate causally. On top of $h_t$, **ATDM** decomposes the evidence-aligned response-timing decision ($t_r = \min\{t|\mathcal{F}(h_t, Q) = A\}$) into a sequence of sub-goals $\mathcal{S}$ and maintains time-indexed tuples $(a_s(t), c_s(t), \rho_s(t))$—sub-answer $a$, confidence $c$, and progress $\rho$—to quantify reasoning and expose rationales. $\mathcal{F}$ denotes decision function, $A$ denotes answer for user question $Q$. ATDM returns the final response time $t_r$ when each sub-goal $s \in \mathcal{S}$ is solved.

### 3.1 HIERARCHICAL PROGRESSIVE SEMANTIC INTEGRATION (HPSI)

To address the goal—*progressive, causal understanding* of the ever-expanding visible set $\mathbb{V}_t$—we introduce **Hierarchical Progressive Semantic Integration** (HPSI). HPSI equips the model with a compact, relation-preserving *cognition state* that is *advanced* as new clips arrive. Concretely, we insert a small number of learnable *aggregation tokens* $\boldsymbol{p}$ at multiple depths and enforce structured sparsity so that evidence is aggregated locally, integrated hierarchically, and propagated causally.

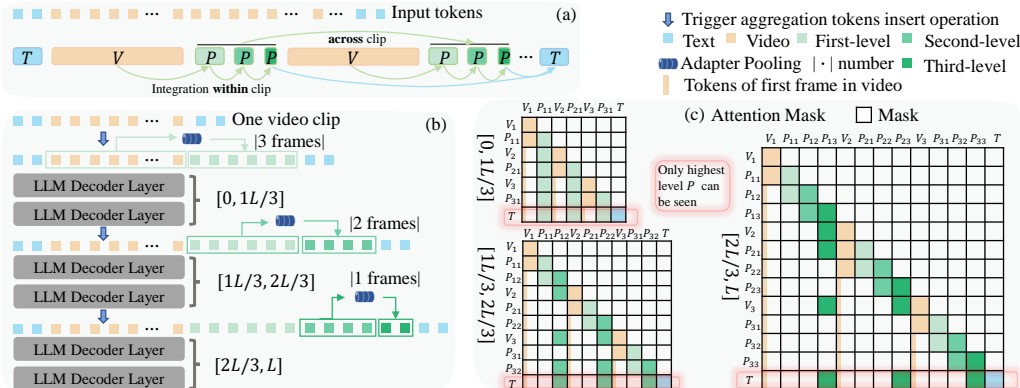

Figure 2: **(a) Visual information aggregation flow diagram. (b) The dynamic integration operation in LLM with a single clip as an example.** The aggregation tokens are initialized in layer 1, layer $1L/3$ and layer $2L/3$ according to the aggregation tokens of the previous level that can support dynamic resolution style, and these tokens are passed forward layer by layer within the LLM to aggregate the visual information of the clip by the causal ability of the LLM and the coefficient attention mask constructed in (c). **(c) Attention mask and its changes.**

**Dynamic-Resolution Progressive Integration Overview.** We segment the video into $n$ clips and, for each $\text{clip}_i$, append a dynamic number of aggregation tokens after its visual tokens. These tokens summarize the semantic content of each clip while leveraging the causal reasoning capabilities of LLMs. Let the original input be $\mathcal{I} = \text{concat}(\boldsymbol{w}, \boldsymbol{v}, \boldsymbol{w})$, with text tokens $\boldsymbol{w}$ and visual tokens $\boldsymbol{v} = (\boldsymbol{v}_{\text{clip}_1}, \ldots, \boldsymbol{v}_{\text{clip}_n})$. We introduce three aggregation levels $j \in \{1, 2, 3\}$, progressively inserted at transformer depths $\ell_j \in \{0, L/3, 2L/3\}$ with target token ratios $r_j \in \{3\times, 2\times, 1\times\}$. For $\text{clip}_i$, level-$j$ produces $n_j(i)$ tokens $\boldsymbol{p}_{\text{clip}_i}^{(j)}$. After inserting the last (level-3) aggregation tokens, the input sequence becomes

$$\widetilde{\mathcal{I}} = \text{concat}\big(\boldsymbol{w}, \boldsymbol{v}_{\text{clip}_1}, \boldsymbol{p}_{\text{clip}_1}^{(1)}, \boldsymbol{p}_{\text{clip}_1}^{(2)}, \boldsymbol{p}_{\text{clip}_1}^{(3)}, \ldots, \boldsymbol{v}_{\text{clip}_n}, \boldsymbol{p}_{\text{clip}_n}^{(1)}, \boldsymbol{p}_{\text{clip}_n}^{(2)}, \boldsymbol{p}_{\text{clip}_n}^{(3)}, \boldsymbol{w}\big). \quad (1)$$

**Aggregation Tokens Initialization.** Each aggregation token is initialized via adaptive average pooling over its clip's visual tokens; let $j = 1, 2, 3$ denote the aggregation level, $\boldsymbol{p}_{\text{clip}_i}^{(0)} = \boldsymbol{v}_{\text{clip}_i}$, and $N_{vc}$ denote the final level's token count (adjustable to match different video resolutions):

$$\boldsymbol{p}_{\text{clip}_i}^{(j)} = \text{AdapterPool}\Big(\boldsymbol{p}_{\text{clip}_i}^{(j-1)}, (4-j)\, N_{vc}\Big), \quad (2)$$

where $\boldsymbol{v}_{\text{clip}_i} \in \mathbb{R}^{n_v \times d}$ represents the $n_v$ visual tokens of the $i$-th clip, and $\text{AdapterPool} : \mathbb{R}^{n_v \times d} \rightarrow \mathbb{R}^{n_c \times d}$ outputs $n_c = (4 - j)N_{vc}$ tokens of dimension $d$.

To guide the model to integrate visual information into these tokens, we construct sparse, structured attention masks (see Fig. 2) that enforce hierarchical visibility: each level-$j$ aggregation token attends only to the preceding level's tokens, ensuring directional semantic consolidation. Text tokens attend causally only to the *last-level aggregation tokens* at each layer. Additionally, we retain visibility for the first-frame tokens of each clip to preserve crucial anchor cues.

**Progressive Integration.** Unlike single-layer average pooling (e.g., LongVA (Zhang et al., 2024b)), HPSI exploits decoder *depth* $L$ by assigning different aggregation strengths across three layer groups: 1) layers $[0, 1L/3]$ integrate raw visual tokens; 2) layers $[1L/3, 2L/3]$ integrate the previous level's tokens; and 3) layers $[2L/3, L]$ refine high-level semantics. With token ratios $3 : 2 : 1$, information is gradually condensed into fewer, more meaningful tokens. Let $\mathcal{L}_j = \{0, L/3, 2L/3\}$,

$$\widetilde{\mathcal{I}}^{(\ell)} = \text{concat}\Big(\boldsymbol{w}, \big(\boldsymbol{v}_{\text{clip}_i}, (\boldsymbol{p}_{\text{clip}_i}^{(k)})_{k=1}^{m(\ell)}\big)_{i=1}^{n}, \boldsymbol{w}\Big), \quad m(\ell) = 1 + \left\lfloor \frac{3\ell}{L} \right\rfloor, \ell \in \mathcal{L}_j, \quad (3)$$

where $n$, $\ell$, and $m(\ell)$ denote the number of clips, the layer index that triggers insertion, and the highest visible aggregation level per clip at layer $\ell$. The output $\boldsymbol{h}_l$ at layer $l \in \{1, ..., L\}$ is

$$\boldsymbol{h}_l = \text{TransformerBlock}\big(\widetilde{\mathcal{I}}^{(\ell)} \odot \mathbb{I}_{l \in \mathcal{L}_j} + \boldsymbol{h}_{l-1} \odot (1 - \mathbb{I}_{l \in \mathcal{L}_j})\big), \quad (4)$$

where $\mathbb{I}_{l \in \mathcal{L}_j}$ is 1 when $l \in \mathcal{L}_j$ and 0 otherwise.

Finally, the progressive integration objective in the semantic space of LLM can be defined as:

$$\min \mathcal{T}_{\text{integration}} = \sum_{l=0}^{L-1} \sum_{j=1}^{3} \Big( \big\| \boldsymbol{p}_{\text{clip}_i}^{(j)(l)} - \text{Pool}\big(\boldsymbol{v}_{\text{clip}_i}\big) \big\|_2 + \big\| \boldsymbol{p}_{\text{clip}_i}^{(j)(l)} - \boldsymbol{p}_{\text{clip}_i}^{(j-1)(l)} \big\|_2 \Big), \quad (5)$$

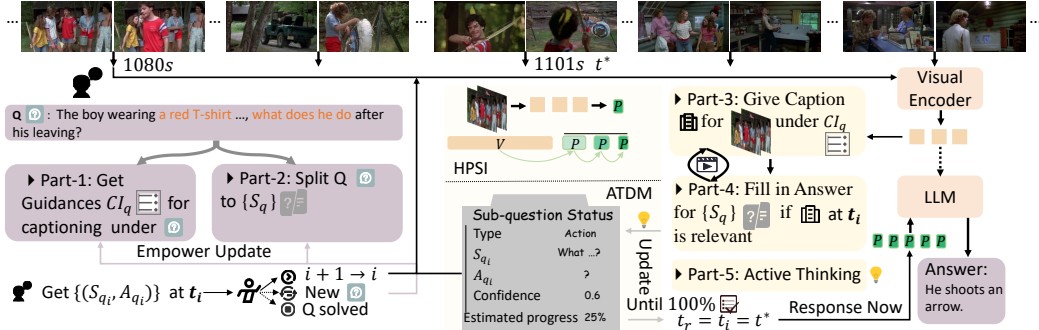

Figure 3: **Pipeline of Thinking-QwenVL.** Given streamed clips and a query $Q$, ATDM generates question-guided caption instructions, decomposes $Q$ into sub-questions, and iteratively extracts evidence from each clip (with progressive visual integration using HPSI), updating sub-answers with progress $\rho \in [0,1]$ and confidence $\mathbf{c} \in [0,1]$. This process runs in parallel across clips and permits to trigger active reflection according to $\mathbf{c}$. The model emits an answer at $t_r = t_i$ once $\boldsymbol{\rho}(t_i) = \mathbf{1}$.

which encourages faithful integration toward clip evidence and smooth refinement across levels.

**Layer-wise Task Decomposition for Hierarchical Aggregation.** Conceptually, HPSI treats the depth of a transformer as a *division of labor* for aggregation, rather than a single pooling step. We explicitly assign different aggregation roles to different layer segments: shallow layers focus on preserving fine-grained local evidence, middle layers consolidate mid-range temporal and structural patterns, and deep layers perform strong semantic condensation into a compact set of high-level summary tokens. In this view, the three levels are not ad-hoc tricks but a progressive aggregation pipeline that incrementally transforms dense visual streams into a small, semantically rich state.

## 3.2 ACTIVE THINKING DECISION MAKER (ATDM)

**Overview.** ATDM converts online answering into a compact, observable chain-of-thought that carries explicit telemetry. Given a streamed video (segmented into clips) and query $q$, ATDM (i) derives question-guided caption requirements and produces a per-clip summary, (ii) decomposes $q$ into $K$ concrete sub-questions, (iii) extracts and updates sub-answers with per-step progress $\rho \in [0,1]$ and confidence $c \in [0,1]^K$, and (iv) declares readiness and answers when all required sub-answers are *confidently* resolved—thereby aligning $t_r$ with the first-sufficient evidence $t^\star$. A modular `wrapper` schedules the per-clip evidence extraction and sub-answer updates in parallel across consecutive clips (e.g., $\text{clip}_i, \text{clip}_{i+1}, \text{clip}_{i+2}$), reducing idle time and preserving responsiveness.

**Active, Self-triggered Thinking.** Beyond the fixed CoT flow, ATDM monitors $\rho$ and $c$ over time; when confidence remains low or the stream exhibits major semantic shifts, it *self-triggers* reflection that revisits prior summaries, constructs cross-clip causal links, and revises hypotheses and metrics $(\rho, c)$. This mechanism prevents myopic updates, improves evidence alignment, and yields more accurate, timely responses under evolving visual evidence.

Combining the above ideas, the **Five-Part Chain-of-Thought Active Thinking Decision Maker Process (ATDM)** is as follows. **Only** Parts 3 and 4 require reasoning to be processed iteratively across video clips rather than forcing all five parts to be executed sequentially for each clip.
▶ Part-1: **Question-Guided Captioning instructions.** Unlike general video captioning models that produce either overly generic descriptions (e.g., "*a person is cooking in a cluttered kitchen*") or irrelevant ones due to unaligned attention, we first ask the model to analyze the question and generate its own captioning guidelines, termed *caption instructions $CI_q$*. These instructions focus the captioning process on questioning-relevant elements.

> **Task:** Analyze the user's question and define exact observation requirements for video captioning to help answer it. Output: [<|Caption Requirements List|>]

▶ Part-2: **Question Decomposition.** Inspired by the progression of human-like reasoning, where answering complex questions involves progressively addressing multiple semantic dimensions, we decompose the original question into a set of sub-questions $\{S_q\}$. These sub-questions structure the reasoning process and allow us to quantify decision progress.

> **Task:** Break the user's question down into a set of precise, concrete sub-questions. Each sub-question can focus on an observable aspect of the video (e.g., object, person, action, spatial

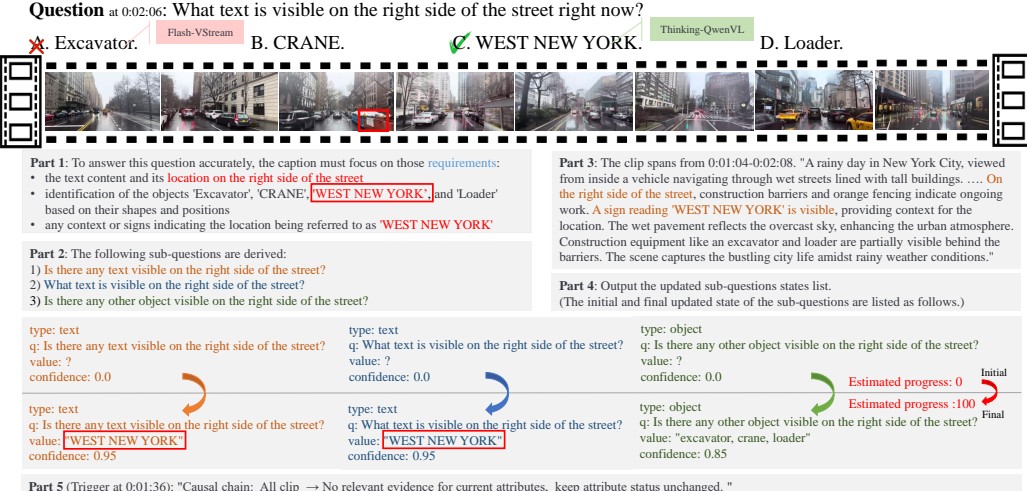

Figure 4: Visualization of qualitative example showcasing how our ATDM framework achieves successful decision and video reasoning.

relation, etc.). These sub-questions represent the key things that must be visually or aurally verified in the video to answer the main question. Output: `[<|Required Subquestions|>]`

▶ **Part-3: Video Clip Captioning.** Based on the *caption instructions* $CI_q$ from part-1, the model generates a summary $\{C_q\}$ of the clip content. This streaming captioning continues until the model determines that sufficient information is available to answer the question.

> **Task:** Watch the current video clip and generate a descriptive caption, you must focus your caption on the following key observation points: `<|Caption Requirements List|>`
> Output: `<|detailed caption that fulfills the requirements|>`

▶ **Part-4: Sub-answer Extraction and Filling.** Using $\{S_q\}$ and the current clip caption $\{C_q\}$, the model attempts to answer each sub-question, forming a set of partial answers $\{SA_q\}$. At each time step, the most recent $\{SA_q\}$ is fed back into the model, enabling it to track historical answer states across frames. This is crucial for effective task decomposition, as corroborated in (Jang et al., 2025).

> **Task:** Read `[Question]`, `[<|Required Subquestions|>]` and the caption of the current video clip `[<|Caption|>]`. For each subquestion, determine whether the caption provides enough information to answer it: - If yes: provide an appropriate answer and a confidence score $c \in [0, 1]$. - If no or uncertain: set value is '?' and $c = 0.0$.
> Output: `<|Updated subquestion state (value,c) and progress ρ|>`

▶ **Part-5: Active Thinking Trigger: Low Confidence or Major Shifts.** Rigid step-by-step reasoning can lead to tunnel vision, causing the model to miss globally coherent information and the relationships between continuously changing information. To mitigate this, we monitor confidence scores for each $\{SA_q\}$. When scores exhibit sharp drops or remain low across time, the model triggers active thinking: it reviews prior $\{C_q\}$, detects temporal shifts, constructs causal chains across clips, and re-evaluates sub-answers accordingly.

> **Task:** Given `[question]` Past reasoning state: `[past cot state]` and the `[new clip caption]`, 1) Cross-clip causal reasoning. Build an explicit, ordered chain that shows how evidence from each new clip supports, contradicts, or refines the current hypothesis. 2) Consistency check. Detect attributes that are contradicted, supported with higher certainty, still low-confidence ($\leq 0.50$), or missing. 3) Update the attribute list and return. Output: `<|Updated subquestion state and progress|>`

**History-aware Decision Process.** The explicit progress and confidence signals $(\rho, c)$ transform ATDM from a sequence of memoryless binary decisions into a genuinely history-aware control process. Each judgment about whether the current visual evidence is "sufficient" is not an isolated yes/no query. Treating it as a mere collection of independent binary decisions breaks the information chain. In contrast, ATDM does not repeatedly decide "answer or wait" based only on the current visual chunk; it *observes* its own past decisions and scores and can refine or revise them as additional evidence arrives. The continuous pair $(\rho, c)$ therefore carries substantially higher information content in context than a single $0/1$ gate, as it not only encodes a bare "stop/continue" signal but also compresses the entire history of intermediate judgments into a compact quantitative state.

Table 1: Accuracy (100%) comparison on StreamingBench focusing on Real-Time Visual Understanding tasks. † indicates the reproduced results. The meaning of each subtask is in Appendix A.5.

| Model | Size | Frames | Pub | Subtasks | | | | | | | | | | |
|---|---|---|---|---|---|---|---|---|---|---|---|---|---|---|
| | | | | OP | CR | CS | ATP | EU | TR | PR | SU | ACP | CT | **All** |
| **Human** | – | – | – | 89.47 | 92.00 | 93.60 | 91.47 | 95.65 | 92.52 | 88.00 | 88.75 | 89.74 | 91.30 | 91.46 |
| *Proprietary MLLMs* | | | | | | | | | | | | | | |
| Gemini 1.5 pro | – | 1 fps | – | 79.02 | 80.47 | 83.54 | 79.67 | 80.00 | 84.74 | 77.78 | 64.23 | 71.95 | 48.70 | 75.69 |
| GPT-4o | – | 64 | – | 77.11 | 80.47 | 83.91 | 76.47 | 70.19 | 83.80 | 66.67 | 62.19 | 69.12 | 49.22 | 73.28 |
| Claude 3.5 Sonnet | – | 20 | – | 73.33 | 80.47 | 84.09 | 82.02 | 75.39 | 79.53 | 61.11 | 61.79 | 69.32 | 43.09 | 72.44 |
| *Open-source Offline Long Video LLMs* | | | | | | | | | | | | | | |
| Video-LLaMA2 | 7B | 32 | ARXIV24 | 55.86 | 55.47 | 57.41 | 58.17 | 52.80 | 43.61 | 39.81 | 42.68 | 45.61 | 35.23 | 49.52 |
| VILA-1.5 | 8B | 14 | ARXIV25 | 53.68 | 49.22 | 70.98 | 56.86 | 53.42 | 53.89 | 54.63 | 48.78 | 50.14 | 17.62 | 52.32 |
| Video-CCAM | 14B | 96 | ARXIV24 | 56.40 | 57.81 | 65.30 | 62.75 | 64.60 | 51.40 | 42.59 | 47.97 | 49.58 | 31.61 | 53.96 |
| LongVA | 7B | 128 | ARXIV24 | 70.03 | 63.28 | 61.20 | 70.92 | 62.73 | 59.50 | 61.11 | 53.66 | 54.67 | 34.72 | 59.96 |
| InternVL-V2 | 8B | 16 | ARXIV24 | 68.12 | 60.94 | 69.40 | 77.12 | 67.70 | 62.93 | 59.26 | 53.25 | 54.96 | 56.48 | 63.72 |
| Kangaroo | 7B | 64 | ARXIV24 | 71.12 | 84.38 | 70.66 | 73.20 | 67.08 | 61.68 | 56.48 | 55.69 | 62.04 | 38.86 | 64.60 |
| LLaVA-NeXT-Video | 32B | 64 | BLOG24 | 78.20 | 70.31 | 73.82 | 76.80 | 63.35 | 69.78 | 57.41 | 56.10 | 64.31 | 38.86 | 66.96 |
| MiniCPM-V-2.6 | 8B | 32 | ARXIV25 | 71.93 | 71.09 | 77.92 | 75.82 | 64.60 | 65.73 | 70.37 | 56.10 | 62.32 | 53.37 | 67.44 |
| LLaVA-OneVision | 7B | 32 | CVPR25 | 80.38 | 74.22 | 76.03 | 80.72 | 72.67 | 71.65 | 67.59 | 65.45 | 65.72 | 45.08 | 71.12 |
| Qwen2.5-VL | 7B | 1fps | ARXIV24 | 78.32 | 80.47 | 78.86 | 80.45 | 76.73 | 78.50 | 79.63 | 63.41 | 66.19 | 53.19 | 73.68 |
| Offline-Long VLLMs Avg | – | – | – | 62.78 | 62.75 | 65.18 | 65.17 | 60.73 | 59.52 | 54.31 | 51.36 | 53.28 | 41.52 | 53.78 |
| *Open-source Online Video LLMs* | | | | | | | | | | | | | | |
| Flash-VStream | 7B | – | ICCV25 | 25.89 | 43.57 | 24.91 | 23.87 | 27.33 | 13.08 | 18.52 | 25.20 | 23.87 | 48.70 | 23.23 |
| VideoLLM-online | 8B | 2fps | CVPR24 | 39.07 | 40.06 | 34.49 | 31.05 | 45.96 | 32.40 | 31.48 | 34.16 | 42.49 | 27.89 | 35.99 |
| Dispider | 7B | 1fps | CVPR25 | 74.92 | 75.53 | 74.10 | 73.08 | 74.44 | 59.92 | 76.14 | 62.91 | 62.16 | 45.80 | 67.63 |
| **Thinking-QwenVL (Ours)** | 7B | 1fps | | 70.27 | 66.67 | 80.00 | 77.97 | 79.31 | 68.66 | 78.26 | 68.18 | 72.31 | 52.38 | **71.60** +3.97↑ |
| Flash-VStream† | 7B | 1fps | ICCV25 | 24.52 | 21.53 | 21.45 | 19.00 | 26.42 | 26.56 | 22.22 | 22.36 | 21.45 | 24.35 | 22.53 |
| Flash-VStream **+ATDM** | 7B | 1fps | ICCV25 | 28.53 | 27.34 | 24.68 | 26.45 | 31.01 | 27.00 | 25.00 | 24.90 | 27.64 | 26.60 | 26.58 +4.05↑ |

| Model | ACR | FPD | Real. | Back. | Forw. | Overall |
|---|---|---|---|---|---|---|
| Human Agents | 92.6 | 91.1 | 93.2 | 92.3 | 92.9 | 92.8 |
| Gemini 1.5 Pro | 67.0 | 68.3 | 70.8 | 62.3 | 57.2 | 65.3 |
| LLaVA-NeXT-Video-7B | 59.6 | 72.3 | 63.3 | 41.7 | 54.2 | 53.1 |
| LLaVA-OneVision-7B | 58.7 | 71.3 | 62.8 | 45.0 | 50.9 | 52.9 |
| Qwen2-VL-7B | 53.2 | 66.3 | 60.7 | 48.6 | 48.9 | 52.7 |
| LongVU-7B | 49.5 | 68.3 | 57.4 | 39.5 | 48.5 | 48.5 |
| *Open-source Online Video-LLMs* | | | | | | |
| Flash-VStream-7B | 32.1 | 29.7 | 29.9 | 25.4 | 44.2 | 33.2 |
| VideoLLM-online-8B | 23.9 | 45.5 | 20.8 | 17.7 | | |
| Dispider-7B | 49.5 | 61.4 | 54.5 | 36.1 | 34.7 | 41.8 |
| Ours (↓ 93.75%) | 54.9 | 67.5 | 55.8 | 47.4 | 28.6 | 46.9 |
| TimeChat-Online-7B (100%) | 46.8 | 69.3 | 61.9 | 41.7 | 36.7 | 46.7 |
| Ours (100%) | **57.2** | **75.0** | **64.7** | 44.3 | **37.6** | **52.5** |

Table 2: Accuracy on **OVOBench**. Real.: Real-Time Visual Perception, Back.: Backward Tracing, Forw.: Forward Active Responding. –: The specific requirements of Forw. task resulted in VideoLLM-online not being able to response in demanded format.

Figure 5: Accuracy improvements over our baseline model on sub-tasks of the **RTVBench** under the same experimental conditions as the RTVBench paper. The overall accuracy of our model increased from 32.75% to 35.87%.

## 4 EXPERIMENT

### 4.1 IMPLEMENTATION DETAILS

**Training Details.** Our model is built upon the Qwen2.5-VL-7B architecture and all experiments are conducted using $4\times$ A100-80G GPUs. The learning rate is set to $2 \times 10^{-6}$, and the model is configured with a maximum input resolution of $448 \times 448$. More hyperparameter details can be found in Appendix A.3. In our setting, the number of clip frames is 32, the number of the three aggregation tokens is 3, 2, and 1 frames's tokens of the videos. So, the number of the final aggregation tokens can be changed with the video resolution setting.

**Benchmarks.** Our evaluation spans complementary *online* and *offline long-video* QA suites that jointly stress real-time perception, temporal alignment, and long-horizon reasoning. **Streaming-Bench** (Lin et al., 2024) targets low-latency, timestamped queries under streaming constraints. **OVOBench** (Niu et al., 2025) enforces *answer-when-ready* timing—models defer responses until sufficient future evidence (real-time perception, forward tracking, active responding). **RTVBench** and **OVBench** (Xun et al., 2025; Huang et al., 2024) probe continuous perception and online spatio-temporal reasoning via multi-timestamp, hierarchical questions and Past/Current/Future anchoring. For offline long-form understanding, **VideoMME**, **MLVU**, **LongVideoBench**, and **LVBench** (Fu et al., 2024; Zhou et al., 2024; Wu et al., 2024; Wang et al., 2024b) cover short clips to hour-long

Table 3: **Accuracy on offline long-video benchmarks**: MLVU, LongVideoBench, VideoMME (w/o subtitles), and LVBench. Videos are divided into 16-frame clips in HPSI (↓93.75% signifies 93.75% reduction in video frames) and up to 256 frames are sampled per video. "100%" for TimeChat-Online (based on Qwen2.5-VL) denotes no dropping—only model parameters; we reproduce this setting in the last row. "100%"(ours) indicates no insertion—only attention redistribution.

| Model | | Frames | MLVU | LongVideoBench | VideoMME | | LVBench |
| --- | --- | --- | --- | --- | --- | --- | --- |
| | | | | | Overall | Long | |
| **Video Length** | | - | 3∼120 min | 8 sec∼60 min | 1∼60 min | 30∼60 min | 30∼120 min |
| **Open-source Offline Video LLMs** | | | | | | | |
| LLaMA-VID-7B | [ECCV24] | 1fps | 33.2 | - | - | - | 23.9 |
| MovieChat-7B | [CVPR24] | 2048 | 25.8 | - | 38.2 | 33.4 | 22.5 |
| LLaVA-NeXT-Video-7B | [BLOG24] | 32 | - | 43.5 | 46.6 | - | 32.2 |
| VideoChat2-7B | [CVPR24] | 16 | 47.9 | 39.3 | 39.5 | 33.2 | 32.5 |
| LongVA-7B | [ARXIV24] | 128 | 56.3 | - | 52.6 | 46.2 | 35.7 |
| Kangaroo-7B | [ARXIV24] | 64 | 61.0 | 54.2 | 56.0 | 46.6 | 39.4 |
| Video-XL-7B | [CVPR25] | 128 | 64.9 | - | 55.5 | 49.2 | - |
| Qwen2.5-VL-7B | [ARXIV25] | 1fps | 66.9 | 61.5 | 63.2 | 50.4 | 43.1 |
| VISTA-7B | [CVPR25] | - | 62.1 | 53.1 | 55.5 | 49.2 | 39.0 |
| **Open-source Online Video LLMs** | | | | | | | |
| Dispider-7B | [CVPR25] | 1fps | 61.7 | - | 57.2 | - | - |
| VideoChat-Online-8B | [CVPR25] | 2fps | - | - | 52.8 | 44.9 | - |
| **Thinking-QwenVL** | | 1fps (↓93.75%) | 59.6 | - | 56.3 | 49.1 | - |
| TimeChat-Online-7B | [ACM25] | 1fps (100%) | 62.6 | 55.4 | 62.4 | 48.4 | - |
| Δ - Qwen2.5-VL | | - | -4.5 | -6.1 | -0.8 | -1.6 | - |
| **Thinking-QwenVL** | | 1fps (100%) | **68.3** | **62.0** | **67.7** | **56.4** | **43.6** |
| Δ - Qwen2.5-VL | | - | **+1.4** | **+0.5** | **+4.5** | **+6.0** | **+0.5** |

videos, emphasizing granular recall and cross-scale reasoning. We follow official scoring protocols (per-suite QA accuracy and aggregates); full task/metric definitions are in Appendix §A.5.

**Comparative Models. 1) Proprietary Assistants.** For completeness, the strong closed-source models as upper-bound references are included: GPT-4o (OpenAI, 2024), Gemini 1.5 Pro (Team et al., 2023), and Claude 3.5 Sonnet (Anthropic, 2024). **2) Offline Long-Video MLLMs.** We compare to the SOTA long-context video understanding models: Video-LLaMA2 (Cheng et al., 2024), VideoChat2 (Li et al., 2024b), Video-CCAM (Fei et al., 2024), VILA-1.5 (Lin et al., 2023), LLaMA-VID (Li et al., 2025), LongVA (Zhang et al., 2024b), Kangaroo (Liu et al., 2024b), MiniCPM-V-2.6 (Yao et al., 2024) and Video-XL (Shu et al., 2024), along with commonly reported baselines ( LLaVA-OneVision (Li et al., 2024a), LLaVA-NeXT-Video (Liu et al., 2024a), InternVL-V2 (Chen et al., 2024c), Qwen2.5-VL (Wang et al., 2024a)). **3) Online Video LLMs.** Online methods include VideoLLM-online (Chen et al., 2024a), Flash-VStream (Zhang et al., 2024a), Dispider (Qian et al., 2025), and TimeChat(-Online) (Ren et al., 2024).

## 4.2 MAIN RESULTS

**StreamingBench.** In Table 1, we compare our model with recent state-of-the-art systems, including Dispider. Our model achieves an accuracy of **71.60%**, setting a new benchmark for this task. Compared to previous online and streaming models, we have improved the state-of-the-art performance by **3.97%** on the online task, increasing the accuracy from Dispider's **67.63%** to our **71.60%**. Furthermore, we also evaluated the effectiveness of our ATDM approach on models without decision-making capabilities, such as Flash-VStream and Qwen2.5-VL. The results indicate that, in the case of Flash-VStream, the model's accuracy increased from **22.53%** to **26.58%**, representing an improvement of **4.01%**. This demonstrates the general applicability of our decision-making method for online video understanding.

**OVOBench, RTVBench, and OVBench.** In Table 2, we compare our proposed method, Thinking-QwenVL, with existing models on OVOBench. Compared to Flash-VStream, which lacks decision-making capabilities (**33.2%**), and Dispider, which incorporates binary opaque decision-making (**41.8%**), our model achieves an accuracy of **46.9%**, marking an improvement of **4.9%** over Dispider. Compared to our baseline, the overall accuracy of our model increased on RTVBench from 32.75% to 35.87%. We achieved 35.6% accuracy on OVBench. The performance on sub-tasks is in Fig. 5 and Table 6& 7. The meaning of each symbol in Fig. 5 is: TP - Temporal Perception, VP - Visual Perception, SP - Scene Perception, PU - Phenomenological Understanding, GU - Global Understanding, IA - Intent Analysis, FP - Faithfulness Prediction, SR - Similarity Reasoning.

Table 4: Impact of **3 level aggregation** on VideoMME w/o subs and OVOBench. We ablate by directly removing the corresponding level tokens. ♠ denotes that the first-stage compressed-token count is set as the final token budget (1×)—equivalent to applying adaptive pooling to visual tokens before the LLM, as in prior long-video models. FF: First Frame. LV: Level. ■ : burden of tokens.

| FF | LV-1 | LV-2 | LV-3 | ■ | OVOBench | | | | VideoMME | | | | AVG |
|---|---|---|---|---|---|---|---|---|---|---|---|---|---|
| | | | | | Overall | Real. | Back. | Forw. | Overall | Short | Medium | Long | |
| ✓ | ✓ | ✓ | ✓ | 1× | 46.9 | 55.8 | 47.4 | 28.6 | 56.3 | 66.0 | 53.9 | 49.1 | 51.6 |
| ✓ | ✓ | ✓ | ✗ | 2× | 46.0 | 53.2 | 48.5 | 29.1 | 56.0 | 65.6 | 53.6 | 49.0 | 51.0 |
| ✓ | ✓ | ✗ | ✗ | 3× | 49.6 | 56.7 | 53.9 | 31.4 | 54.7 | 61.9 | 54.7 | 47.6 | 52.2 |
| ✗ | ✓ | ✓ | ✓ | - | 42.6 | 49.1 | 42.2 | 30.2 | 49.7 | 55.9 | 48.6 | 44.7 | 46.2 |
| ✓ | ♠ | ✗ | ✗ | 1× | 43.4 ↓**3.5** | 45.4 | 52.7 | 29.9 | 48.9 ↓**7.4** | 52.4 | 49.0 | 45.1 | 46.2 |

Figure 6: The impact of **ATDM** on OVO-Bench and StreamingBench subtasks.

Figure 7: Impact of **ATDM components**. All represents the complete model performance when use ATDM. Each column beyond this represents the ablation of the corresponding part of ATDM.

**VideoMME and MLVU.** Although our model is optimized for online scenarios, it still demonstrates competitive performance on long-video benchmarks. Our model achieves **56.3%** on VideoMME, **49.1%** on VideoMME-Long, and **61.2%** on MLVU, outperforming several models specifically designed for offline long-video understanding. When the experimental setup is configured to use only the modified attention weight distributions (100%), the accuracy reaches **68.3%** on MLVU, **62.0%** on LongVideoBench, **67.7%** on VideoMME, and **43.6%** on LVBench, surpassing existing state-of-the-art offline long-video models. Notably, on VideoMME-Long ($30 \sim 60$ min), it outperforms the leading Qwen2.5-VL-7B by **6%** in accuracy. This strongly demonstrates the effectiveness of our HPSI module for video understanding, as this progressive causal approach that incrementally enhances the model's cognitive state proves effective for tasks requiring long-term dependencies.

## 4.3 ABLATION STUDY

**Overview.** We conduct a comprehensive ablation study in two dimensions: 1) the impact of hierarchical integration across different layers, and 2) the contribution of each part in ATDM.

**HPSI and Three-Level Aggregation Tokens.** Table 4 ablates the per-level insertions of HPSI. A salient finding is that removing levels 2–3 and forcing level 1 (the first LLM layer) to downsample directly to the same token budget as our level-3 setting—i.e., a *single-shot AdapterPooling* baseline applied *before* the LLM—reduces accuracy by **3.5%** on OVOBench and **7.4%** on VideoMME-Long. This confirms that one-stage pooling discards fine-grained cues and disrupts long-range, cross-clip dependencies; HPSI cannot be replaced by simple pooling. On offline long-video benchmarks (Table 3), Thinking-QwenVL further surpasses the baseline by **4.5%** on VideoMME, and—under the same backbone and comparable data coverage—outperforms TimeChat-Online by **5.9%** on MLVU, **6.6%** on LongVideoBench, and **7.6%** on VideoMME-Long. Together, these results show that HPSI's *multi-depth aggregation tokens* and *structured sparse attention* preserve semantics under tight budgets and enable stronger causal reasoning over extended evidence than single-step pooling.

**ATDM and its Components.** We evaluate the decision-making capability of ATDM across three models on OVOBench and StreamingBench, as shown in Fig. 6. Models without decision-making capabilities show significant performance improvements with ATDM. For example, on the OVOBench-EPM sub-task, all three models achieve more than a **5%** accuracy boost. In Table 1, Flash-VStream's performance on StreamingBench increases from **22.53%** to **26.58%**, a **4.05%** gain. These results demonstrate that streaming and offline video understanding models, when operating under paradigms like $t_r = t_q$ or $t_r = T$, suffer from performance limitations. However, when equipped with decision-making capabilities aligned with visual evidence, model accuracy significantly improves. We further isolate the contribution of each component ($P_1$-$P_5$) on Thinking-QwenVL and Flash-VStream in Fig. 7. Each part is either removed or replaced with alternative operations. More detailed experimental setups are provided in Appendix A.2.

| Robust | | | |
|---|---|---|---|
| Setting | 100% | 80% | 70% |
| Acc. (%) | 71.60 | 68.75 | 67.81 |
| **Applicability** | | | |
| Framework | F-VStream | Ours | |
| Base Model | LLaVA-7B | QwenVL-3B | QwenVL-7B |
| Acc. (%) | 26.58 | 62.62 | **71.60** |

Table 5: **Robustness and Applicability.** *Top:* Stress-testing streaming robustness under abnormal conditions by uniformly dropping frames *after* 1 FPS extraction (retaining 100%, 80%, 70%). *Bottom:* The ATDM controller is applied to multiple backbones (Flash-VStream-LLaVA-7B, Our Thinking-Qwen2.5-VL-3B/7B), showing its framework-agnostic utility.

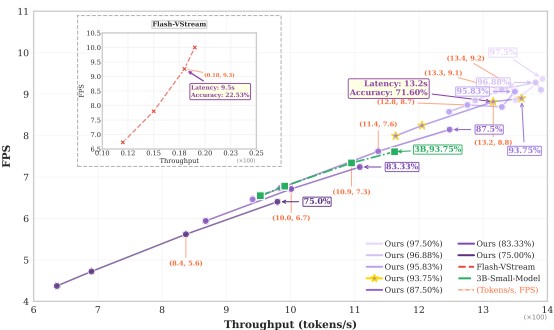

Figure 8: **Efficiency on NVIDIA A100 GPUs.** Impact of the aggregation rate in HPSI on FPS and token throughput. At 93.75% aggregation rate, our method matches Flash-VStream's FPS (8.49 vs. 8.45) with 78× higher avg. token throughput (1261 vs. 16), and a slight latency increase (13.2ms vs. 9.5ms).

**Robustness and Applicability.** We evaluate stability under abnormal streaming conditions (*missing frames* and *abrupt scene transitions*) by starting from the 1 FPS protocol and uniformly retaining only 80% and 70% of frames. As shown in Table 5, the degradation is mild: even at **70%** retention, StreamingBench accuracy remains **67.81%**. Combined with the ablations in Table 4, this indicates that HPSI and ATDM preserve performance under aggressive frame loss. Beyond robustness, the two modules transfer across backbones and frameworks. Instantiating ATDM on the real-time Flash-VStream pipeline (built on LLaVA) consistently activates time-stamped decision making and yields a $\sim 4.0\%$ accuracy gain on StreamingBench (Table 1). Applying HPSI and ATDM to a smaller Qwen2.5-VL-3B backbone, Thinking-QwenVL-3B reaches **62.62%** in Table 5, only about 5% below the 7B Dispider model and far above the 8B VideoLLM-Online model (35.99%). Methodologically, HPSI requires only a deep LLM, which we partition into three segments with increasing aggregation strength, without backbone changes (Fig. 2). ATDM relies only on basic instruction-following and visual comprehension, arousing the decision-making ability of generic VLMs easily.

**Efficiency and Feasibility.** On StreamingBench (NVIDIA A100 GPUs), our 93.75% aggregation setting matches Flash-VStream in frame throughput (8.49 vs. 8.45 FPS) while yielding higher accuracy (71.60% vs. 22.53%). The explicit decision pipeline adds a modest $\sim 3.7s$ end-to-end latency (13.2s vs. 9.5s). Sweeping aggregation (75%→97.5%) monotonically improves total average FPS (5.28→9.12) and token throughput (tokens/s) (786→1351; scaled ×0.01 in Fig. 8), providing a simple speed–quality knob. A 3B variant at 93.75% still attains average 7.07 FPS / 1050 throughput.

**Qualitative Effect of HPSI on ATDM.** On the painting clip in Fig. 13, HPSI supplies ATDM with a temporally consolidated memory, yielding captions that explicitly **encode state changes over time** (e.g., "*the brush moves from right to left*" and "*the hand adjusts its angle*"), rather than a single, static snapshot. In contrast, the baseline—lacking hierarchical integration—produces short, largely scene-static descriptions with weak cross-frame cohesion. This qualitative gap indicates that **HPSI's multi-level aggregation preserves and stabilizes evolving visual evidence across frames**, which ATDM then leverages to issue timestamped, evidence-aligned decisions; the same synergy remains observable even when frames are missing or hard cuts introduce abrupt scene transitions. We also provide an intuitive comparison of our model and Flash-VStream's output examples in Fig. 11&12.

## 5 CONCLUSION

We introduced Thinking-QwenVL, which integrates Hierarchical Progressive Semantic Integration (HPSI) with an Active Thinking Decision Maker (ATDM). HPSI maintains a compact, relation-preserving cognition state that is progressively updated as evidence accrues under structured sparsity, while ATDM complements this with a decision process that decomposes tasks into observable sub-goals, enriched by progress metrics, confidence estimates, and a readiness head aligned to first-sufficient evidence. Empirical evaluation shows that Thinking-QwenVL achieves strong results on online benchmarks and remains competitive on offline long-video tasks, with ablations confirming that HPSI's multi-depth aggregation and ATDM's decision process are key to both accuracy and timely responses.

ACKNOWLEDGMENTS

This work was partially supported by New Generation Artificial Intelligence-National Science and Technology Major Projection(2025ZD0123100) and by The National Natural Science Foundation of China (NSFC) under no. 62573399 and U25A20530.

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

# A    ADDITIONAL EXPERIMENTAL SETTINGS AND RESULTS

## A.1    MORE RESULTS

To thoroughly showcase the capabilities of Thinking-QwenVL, we provide supplementary experimental data in Table 6& 7& 8, and and attention mask visualization in Fig. 10.

Table 6: **Performance comparison (accuracy** $100\%$**) on OVBench.**  Subtasks in it are AA: Action Anticipation, GSP: Goal/Step Prediction, MP: Movement Prediction, AP: Action Persistence, SV: Step Verification, OP: Object Presence, AR: Action Retrieval, PR: Procedure Recall, TR: Trajectory Retrieval, AL: Action Location, OP: Object Position, AT: Action Trajectory, OT: Object Trajectory, AS: Action Sequence, SL: Step Localization, OES: Object Existence State.

| Task Name | Size | FP | | | THV | | | PM | | | SP | | STP | | TP | | | AVG |
|---|---|---|---|---|---|---|---|---|---|---|---|---|---|---|---|---|---|---|
| | | AA | GSP | MP | AP | SV | OP | AR | PR | TR | AL | OP | AT | OT | AS | SL | OES | |
| *Proprietary Multimodal Models* | | | | | | | | | | | | | | | | | | |
| Gemini-1.5-Flash | - | 71.4 | 53.6 | 21.9 | 56.5 | 60.8 | 40.6 | 36.7 | 47.9 | 62.5 | 32.3 | 37.5 | 87.0 | 50.0 | 83.3 | 22.3 | 46.9 | 50.7 |
| *Open-source Offline Long Video LLMs* | | | | | | | | | | | | | | | | | | |
| InternVL2 | 7B | 52.6 | 60.2 | 27.6 | 57.5 | 52.0 | 58.5 | 38.8 | 67.1 | 58.3 | 38.1 | 31.3 | 87.4 | 37.0 | 75.4 | 31.4 | 5.9 | 48.7 |
| InternVL2 | 4B | 57.7 | 57.0 | 14.4 | 59.2 | 49.4 | 60.0 | 30.3 | 61.8 | 46.3 | 30.9 | 20.1 | 83.0 | 32.3 | 70.7 | 29.4 | 3.4 | 44.1 |
| LLaMA-VID | 7B | 43.6 | 50.9 | 19.6 | 64.0 | 47.5 | 46.8 | 29.4 | 48.9 | 51.2 | 31.9 | 11.2 | 75.7 | 24.8 | 59.1 | 26.0 | 40.0 | 41.9 |
| LLaVA-Onevision | 7B | 68.0 | 62.7 | 35.9 | 58.4 | 50.3 | 46.5 | 29.4 | 60.7 | 58.0 | 43.1 | 14.2 | 86.5 | 49.7 | 70.7 | 28.1 | 30.2 | 49.5 |
| LongVA | 7B | 64.1 | 56.5 | 29.5 | 54.9 | 51.9 | 34.8 | 35.3 | 55.6 | 57.7 | 31.6 | 3.4 | 67.4 | 44.7 | 80.0 | 26.7 | 4.0 | 43.6 |
| MiniCPM-V2.6 | 7B | 33.3 | 35.9 | 15.0 | 59.2 | 50.8 | 55.1 | 25.0 | 37.4 | 41.7 | 26.6 | 11.8 | 98.3 | 36.3 | 66.1 | 26.4 | 6.2 | 39.1 |
| Qwen2-VL | 7B | 60.3 | 66.1 | 22.1 | 54.9 | 51.5 | 51.1 | 37.8 | 64.4 | 69.3 | 35.3 | 28.5 | 97.0 | 49.4 | 65.1 | 30.8 | 11.7 | 49.7 |
| LITA | 7B | 19.2 | 24.5 | 19.9 | 40.8 | 48.9 | 24.9 | 3.1 | 27.3 | 6.4 | 6.9 | 14.6 | 35.2 | 23.9 | 27.4 | 0.5 | 3.4 | 20.4 |
| TimeChat | 7B | 7.7 | 15.3 | 18.7 | 20.6 | 15.7 | 11.7 | 9.1 | 14.7 | 9.8 | 7.5 | 19.5 | 13.9 | 10.3 | 9.3 | 10.1 | 10.8 | 12.8 |
| VTimeLLM | 7B | 37.2 | 23.4 | 15.0 | 64.8 | 43.8 | 53.2 | 25.9 | 38.8 | 32.5 | 25.9 | 20.4 | 40.9 | 6.8 | 48.4 | 43.5 | 8.6 | 33.1 |
| *Open-source Online Video-LLMs* | | | | | | | | | | | | | | | | | | |
| VideoLLM-Online | 7B | 0 | 1.8 | 20.9 | 5.2 | 5.9 | 32.6 | 0 | 2.3 | 26.7 | 0.6 | 26.6 | 0.9 | 19.9 | 0.9 | 1.7 | 8.3 | 9.6 |
| MovieChat | 7B | 23.1 | 27.5 | 23.6 | 58.4 | 43.9 | 40.3 | 25.6 | 31.1 | 23.9 | 26.9 | 39.6 | 24.4 | 28.9 | 29.3 | 25.5 | 21.9 | 30.9 |
| Flash-Vstream | 7B | 26.9 | 37.6 | 23.9 | 60.1 | 41.9 | 40.0 | 23.4 | 35.3 | 26.1 | 24.7 | 28.8 | 27.0 | 21.4 | 29.8 | 25.6 | 26.8 | 31.2 |
| Thinking-QwenVL | 7B | 27.8 | 39.6 | 25.9 | 62.2 | 42.3 | 41.4 | 25.3 | 36.3 | 27.1 | 24.4 | 30.8 | 27.6 | 25.1 | 30.2 | 26.5 | 27.6 | **35.6** +4.4↑ |

Table 7: **Accuracy (**$100\%$**) on RTVBench.**  We evaluate without audio; otherwise, all settings—including the frame-sampling method—follow RTVBench (Xun et al., 2025) for a fair comparison. Compared with our baseline model, the *overall accuracy* of our approach improves from $32.75\%$ to $35.87\%$, yielding a gain of **3.12%**. The Subtasks in it are: Temporal Perception (TP), Visual Perception (VP), Scene Perception (SP), Global Understanding (GU), Phenomenological Understanding (PU), Intent Analysis (IA), Future Prediction (FP), and Spatiotemporal Reasoning (SR).

| Method | Size | TP | VP | SP | GU | PU | IA | FP | SR |
|---|---|---|---|---|---|---|---|---|---|
| *Closed-Source Business Models* | | | | | | | | | |
| Gemini 2.0 Flash | - | 40.49 | 45.19 | 39.34 | 35.70 | 45.65 | 46.78 | 44.42 | 38.46 |
| GPT-4o | - | 48.60 | 53.59 | 52.63 | 45.02 | 54.32 | 48.58 | 54.67 | 42.75 |
| *Open-Source Offline Video Models* | | | | | | | | | |
| VideoLLaMA2 | 7B | 39.52 | 42.49 | 39.85 | 37.34 | 42.21 | 40.92 | 41.47 | 33.50 |
| VideoLLaMA3 | 7B | 37.82 | 39.24 | 36.87 | 33.54 | 39.13 | 33.39 | 38.05 | 33.84 |
| LLaVA-OneVision | 7B | 35.09 | 35.86 | 35.20 | 32.07 | 33.51 | 37.06 | 38.23 | 28.91 |
| LLaVA-Video | 7B | 34.07 | 38.97 | 34.45 | 29.42 | 35.69 | 36.33 | 39.08 | 31.22 |
| Qwen2.5-VL | 7B | 32.37 | 37.48 | 30.73 | 29.11 | 35.69 | 29.36 | 35.33 | 33.67 |
| Ours | 7B | 37.65 +5.3↑ | 41.00 +3.5↑ | 30.17 | 31.86 | 32.66 | 37.86 +8.5↑ | 37.20 | 35.88 |

## A.2    DETAILS ABOUT THE COMPONENTS ANALYSIS OF ATDM

In Fig. 7, we present five sets of ablation experiments on the components of ATDM, conducted on two models. These five sets of experiments are based on the following control conditions:

1) $P_1$: Remove $P_1$ (*caption instructions* $CI_q$); demand $P_2$ give captions directly.

2) $P_2$: Disable $P_2$ *Question decomposition*; retain a single query $Q$ and require $P_4$ to answer $Q$ at each step while still emitting per-step confidence $c$ and progress $\rho$.

3) $P_3$: Remove P3 *Streaming captioning* to test the value of the textual intermediary; $P_4$ is switched from text-only consumption to *multimodal* extraction—directly retrieving evidence from the current visual stream to fill sub-answers.

Table 8: Comparison with current online Video understanding LMMs on **OVOBench**. The subtasks are: i) *Real-Time Visual Perception* (OCR: Optical Character Recognition, ACR: Action Recognition, ATR: Attribute Recognition, STU: Spatial Understanding, FPD: Future Prediction, OJR: Object Recognition), ii) *Backward Tracing* (EPM: Episodic Memory, ASI: Action Sequence Identification, HLD: Hallucination Detection), and iii) *Forward Active Responding* (REC: Repetition Event Count, SSR: Sequential Steps Recognition, CRR: Clues Reveal Responding).

| Model | #Frames | Real-Time Visual Perception | | | | | | | Backward Tracing | | | | Forward Active Responding | | | | Overall |
|---|---|---|---|---|---|---|---|---|---|---|---|---|---|---|---|---|---|
| | | OCR | ACR | ATR | STU | FPD | OJR | Avg. | EPM | ASI | HLD | Avg. | REC | SSR | CRR | Avg. | Avg. |
| Human Agents | - | 94.0 | 92.6 | 94.8 | 92.7 | 91.1 | 94.0 | 93.2 | 92.6 | 93.0 | 91.4 | 92.3 | 95.5 | 89.7 | 93.6 | 92.9 | 92.8 |
| *Proprietary Multimodal Models* | | | | | | | | | | | | | | | | | |
| Gemini 1.5 Pro | 1fps | 87.3 | 67.0 | 80.2 | 54.5 | 68.3 | 67.4 | 70.8 | 68.6 | 75.7 | 52.7 | 62.3 | 35.5 | 74.2 | 61.7 | 57.2 | 65.3 |
| GPT-4o | 64 | 69.1 | 65.1 | 65.5 | 50.0 | 68.3 | 63.7 | 63.6 | 49.8 | 71.0 | 55.4 | 58.7 | 27.6 | 73.2 | 59.4 | 53.4 | 58.6 |
| *Open-source Offline Long Video LLMs* | | | | | | | | | | | | | | | | | |
| LLaVA-NeXT-Video-7B | 64 | 69.8 | 59.6 | 66.4 | 50.6 | 72.3 | 61.4 | 63.3 | 51.2 | 64.2 | 9.7 | 41.7 | 34.1 | 67.6 | 60.8 | 54.2 | 53.1 |
| LLaVA-OneVision-7B | 64 | 67.1 | 58.7 | 69.8 | 49.4 | 71.3 | 60.3 | 62.8 | 52.5 | 58.8 | 23.7 | 45.0 | 24.8 | 66.9 | 60.8 | 50.9 | 52.9 |
| Qwen2-VL-7B | 64 | 69.1 | 53.2 | 63.8 | 50.6 | 66.3 | 60.9 | 60.7 | 44.4 | 66.9 | 34.4 | 48.6 | 30.1 | 65.7 | 50.8 | 48.9 | 52.7 |
| InternVL-V2-8B | 64 | 68.5 | 58.7 | 69.0 | 44.9 | 67.3 | 56.0 | 60.7 | 43.1 | 61.5 | 27.4 | 44.0 | 25.8 | 57.6 | 52.9 | 45.4 | 50.1 |
| LongVU-7B | 1fps | 55.7 | 49.5 | 59.5 | 48.3 | 68.3 | 63.0 | 57.4 | 43.1 | 66.2 | 9.1 | 39.5 | 16.6 | 69.0 | 60.0 | 48.5 | 48.5 |
| *Open-source Online Video-LLMs* | | | | | | | | | | | | | | | | | |
| Flash-VStream-7B | 1fps | 25.5 | 32.1 | 29.3 | 33.7 | 29.7 | 28.8 | 29.9 | 36.4 | 33.8 | 5.9 | 25.4 | 5.4 | 67.3 | 60.0 | 44.2 | 33.2 |
| VideoLLM-online-8B | 2fps | 8.1 | 23.9 | 12.1 | 14.0 | 45.5 | 21.2 | 20.8 | 22.2 | 18.8 | 12.2 | 17.7 | - | - | - | - | - |
| Dispider | 1fps | 57.7 | 49.5 | 62.1 | 44.9 | 61.4 | 51.6 | 54.5 | 48.5 | 55.4 | 34.7 | 4.3 | 36.1 | 18.0 | 37.4 | 48.8 | 41.8 |
| TimeChat-Online-7B | 1fps (100%) | 75.2 | 46.8 | 70.7 | 47.8 | 69.3 | 61.4 | 61.9 | 55.9 | 59.5 | 9.7 | 41.7 | 31.6 | 38.5 | 40.0 | 36.7 | 46.7 |
| Ours | 1fps (↓ 93.75%) | 56.4 | 54.9 | 60.4 | 45.0 | 67.5 | 50.4 | 55.8 | 41.7 | 55.9 | 44.7 | 47.4 | 12.0 | 33.8 | 40.0 | 28.6 | 46.9 |
| Ours | 1fps (100%) | 74.1 | 57.2 | 68.1 | 55.3 | 75.0 | 58.3 | 64.7 | 48.0 | 56.3 | 28.8 | 44.3 | 29.1 | 39.3 | 40.0 | 36.1 | **52.5** |

4) $P_4$: Replace the graded $(\rho, c)$ update in $P_4$ (*Progressive tracking sub-questions status*) with a single binary answerable flag (0/1), eliminating accumulated progress and confidence smoothing.

5) $P_5$: Remove $P_5$ (*self-triggered reflection*) to assess the benefit of cross-clip causal revision under low confidence or major semantic shifts.

## A.3 SUMMARY OF HYPERPARAMETER SETTINGS

The training process of our Thinking-QwenVL is structured into three distinct phases. **1) Integration Pre-training.** We pretrain the model on LLAVA-Video-178k (Li et al., 2024a) and ShareGPT4v-40k (Chen et al., 2024b), both containing caption-style data. This stage enables the model to learn how to aggregate and compress visual information into the inserted compress tokens at specified positions. **2) Integration-Based Time Perception Learning.** We fine-tune the model on TimeChat-Online-139k (Yao et al., 2025), a dataset annotated with binary labels indicating whether a question is answerable at a given timestamp. This trains the model to decide whether the compressed visual information is sufficient for answering, relying solely on the compress tokens. **3) Interaction-Focused QA Fine-Tuning.** We further fine-tune the model using general QA-style dialog data to enhance its interaction ability and improve alignment with user queries in a streaming setting. Throughout all stages, only the intermediate `Merge` layers and the `LLM` backbone are fine-tuned, while the `visual encoder` remains frozen. All experiments are run on A100 GPUs. Table 9 provides a comprehensive overview of the hyperparameter configurations employed during each training stage.

## A.4 POSITION IDS EMBEDDING FOR INTEGRATION

**Impact of Positional Encoding.** The original QwenVL2.5 model adopts a 3D Rotary Position Embedding (3D RoPE) mechanism. When introducing new aggregation tokens, it becomes necessary to redefine their positional encoding. To maintain compatibility with the model's dynamic spatial resolution handling, we insert aggregation tokens in multiples of the original frame tokens. In Thinking-QwenVL, we retain the 3D RoPE format while adjusting the temporal dimension of the inserted aggregation tokens as in Algorithm 1. This ensures the spatial indices are aligned with the original frames while preserving temporal distinction across hierarchical aggregation levels. To evaluate this strategy, we replace 3D RoPE with a sequential positional encoding and introduce a new variant, *Offset Sequential Positional Embedding* (OSPR). OSPR explicitly offsets the sequential position IDs of aggregation tokens according to their hierarchy level. On OVOBench, substituting 3D RoPE with OSPR reduces overall accuracy from $46.9\%$ to $43.3\%$ (a drop of **3.6** percentage points), which is also a reason we retain 3D RoPE in our model.

Table 9: Training hyperparameters of Thinking-QwenVL for all stages.

| Configuration | Integration Pre-training | Time Perception Learning | Interaction-Focused QA Tuning |
|---|---|---|---|
| Training Datasets | LLAVA-Video-178k&ShareGPT4v-40k | TimeChat-Online-139k | LLAVA-Video-178k |
| Training Datasets Type | Caption | Open-ended QA | Multiple-choice QA |
| Training Modules | LLM&Merge Layer | LLM&Merge Layer | LLM&Merge Layer |
| Frame Resolution | $448 \times 448$ | $448 \times 448$ | $448 \times 448$ |
| Max Frames | 128 | 196 | 128 |
| Optimizer | AdamW | AdamW | AdamW |
| Learning Rate | $2e^{-6}$&$1e^{-5}$ | $2e^{-6}$&$1e^{-5}$ | $2e^{-6}$&$1e^{-5}$ |
| Learning Rate Schedule | cosine decay | cosine decay | cosine decay |
| Weight Decay | 0.1 | 0.1 | 0.1 |
| Gradient Clip | 1.0 | 1.0 | 1.0 |
| Warm-up Ratio | 0.03 | 0.03 | 0.03 |
| Global Batch Size | 16 | 16 | 16 |
| Numerical Precision | bfloat16 | bfloat16 | bfloat16 |

---

**Algorithm 1** The algorithm of Position IDs embedding for aggregation tokens.

**Require:** $\mathbf{X}$: Input tokens
**Require:** $\mathcal{G}$: video grid $(T, H, W)$
**Require:** $\mathcal{C}$: compress params $(N_{\text{clips}}, N_{\text{comp}}^{(l)})$
**Require:** $\mathcal{P}$: position params $(\Delta t, \tau, S)$
1: $T_{\text{extended}} \leftarrow T + N_{\text{clips}} \times N_{\text{comp}}^{(l)}$
2: $\mathbf{P}_t \leftarrow [0, 1, \ldots, T_{\text{extended}} - 1] \times \Delta t \times \tau$
3: $\mathbf{P}_h \leftarrow \lfloor [0, 1, \ldots, \lfloor H/S \rfloor - 1] \rfloor$
4: $\mathbf{P}_w \leftarrow \lfloor [0, 1, \ldots, \lfloor W/S \rfloor - 1] \rfloor$
5: $\mathbf{M}_t \leftarrow \text{repeat}(\mathbf{P}_t, \text{ along spatial dims})$
6: $\mathbf{M}_h \leftarrow \text{repeat}(\mathbf{P}_h, \text{ along temporal and width dims})$
7: $\mathbf{M}_w \leftarrow \text{repeat}(\mathbf{P}_w, \text{ along temporal and height dims})$
8: $\mathbf{Pos}_{\text{3D}} \leftarrow \text{stack}(\mathbf{M}_t, \mathbf{M}_h, \mathbf{M}_w)$
9: **return** $\mathbf{Pos}_{\text{3D}}$

---

## A.5 EVALUATION METRICS

**StreamingBench** (Lin et al., 2024) is a large-scale **online** video benchmark spanning *900* videos with *4,500* timestamped multiple-choice QAs, designed to test real-time perception and interaction under realistic stream constraints. Tasks are grouped into three families: *Real-Time Visual Understanding*, *Omni-Source Understanding*, and *Contextual Understanding*. Findings reveal clear gaps: offline long-video MLLMs transfer modestly to real-time *visual* tasks but underperform on *omni-source* and *contextual* tasks requiring audio fusion, long-horizon memory, and event-timed actuation; dedicated streaming models remain immature. Each of the 3 types has a split, and since the other two test tasks are non-visual modality-dominant, e.g., the omni-source subset is dominated by the audio modality, we tested on the first split–Real-Time Visual Understanding (2,500 QAs). The subtasks in it are as follows: Object Perception (OP), Causal Reasoning (CR), Clips Summarization (CS), Attribute Perception (ATP), Event Understanding (EU), Text-Rich Understanding (TR), Prospective Reasoning (PR), Spatial Understanding (SU), Action Perception (ACP), and Counting (CT). We use abbreviations for these subtasks in Table 1.

**OVOBench** (Niu et al., 2025) is a dedicated benchmark designed to evaluate online video understanding models with tasks of 3 types (*real-time visual perception / forward tracking / forward active response*). It comprises *644* videos and around *2800* QA pairs, requiring models to *withhold* an answer until sufficient future evidence arrives. OVOBench specifically evaluates temporal alignment capabilities by enforcing strict separation between the query timestamp and the earliest timestamp at which the question becomes answerable. This is particularly important for assessing whether a model can respond at the right moment based on sufficient and relevant visual evidence.

The suite spans **12 tasks** grouped into three modes: *Backward Tracing*—Episodic Memory (EPM), Action Sequence Identification (ASI), Hallucination Detection (HLD); *Real-Time Visual Perception*—Spatial Understanding (STU), Object/Attribute/Action Recognition (OJR/ATR/ACR), OCR, and Future Prediction (FPD); and *Forward Active Responding*—Repetition Event Count (REC), Scene-State Regression (SSR), and Cautious Response Regulation (CRR).

**RTVBench** (Xun et al., 2025) and **OVBench** (Huang et al., 2024) jointly offer a complementary yardstick for online video understanding—probing *continuous perception* and *online spatiotemporal reasoning* under real-time constraints. **RTVBench** (*552 videos / 4,631 QA pairs*) is built around (i) *Multi-Timestamp QA* and a *Hierarchical Question Structure* to prevent shortcutting that can be summarized into three sub-tasks—*Perception*, *Understanding*, and *Reasoning* (future prediction/spatiotemporal reasoning). **OVBench** (*5,000* QAs) scales *online* evaluation across *6* task types with videos ranging from seconds to one hour; it uniquely anchors each query to *Past/Current/Future* temporal contexts, requires fine-grained grounding. Together, the two benchmarks expose persistent limitations of current MLLMs: offline long-video models lose robustness under cluttered, evolving streams and dedicated online models still trail top proprietary systems—highlighting the need for more advanced architectures.

**VideoMME, MLVU, LongVideoBench and LVBench** (Fu et al., 2024; Zhou et al., 2024; Wu et al., 2024; Wang et al., 2024b) are four long video QA benchmarks. VideoMME (2,700 QA pairs) spans six domains with videos from short clips ($< 4$ min) to long-form ($> 1$ h), testing perception, reasoning, and synopsis across temporal scales. MLVU (1,730 videos / 2,593 QA pairs) ranges from 3 minutes to 2 hours, providing complementary coverage of long-form video understanding. LVBench probes extreme long-video comprehension with videos up to two hours (68 min on average). LongVideoBench (*3,763* videos / *6,678* human-authored QA pairs) is a large-scale benchmark for understanding long contexts, which collectively demand granular recall and spatio-temporal reasoning under long inputs.

### A.6 EVALUATION OF DECISION CLARITY AND RATIONALE CORRECTNESS

To systematically assess whether ATDM genuinely improves users' understanding and trust in the model's behaviour, we design four complementary metrics and apply them to the intermediate reasoning traces produced by each ATDM component as well as to the overall decision process. All four metrics are rated on a 1–5 Likert scale (higher is better), and are used consistently by human experts, trained non-experts, and strong LLM judges (`GPT-4o` and `Qwen2.5-VL-72B`). Together, they disentangle: 1) Reasoning Readability: how well the full reasoning text is written and structured; 2) Decision Transparency: how clearly the timing of "answer" vs. "wait" is explained; 3) $(\rho, c)$ Consistency: whether the explicit progress/confidence signals $(\rho, c)$ behave in a numerically consistent manner; and 4) Rationale Correctness: whether the rationale is factually and causally sufficient to justify the decision. The detailed indicators and the meaning of each corresponding score are in Table 12.

**Reasoning Readability** evaluates whether each part (e.g., caption, sub-question answers) in the ATDM reasoning trace follows the *requested content specification, is easy to read, and is locally coherent*. High scores indicate that each component strictly adheres to the instructions, the content is strongly related to the corresponding sub-question, the caption is fluent and covers the required aspects, and the sub-question answers align naturally with the caption, yielding a globally coherent and well-structured reasoning trajectory.

**Decision Transparency** measures whether a rater can clearly understand *why* ATDM decides to answer or to keep waiting at each step, given access to the reasoning trace, the current sub-task states, and the associated progress/confidence scores $(\rho, c)$. A high transparency score means that the trace explicitly indicates when key evidence appears, how sub-tasks are resolved, how $(\rho, c)$ are updated, and how these factors jointly trigger the timing of the decision.

$(\rho, c)$ **Consistency** focuses specifically on the numerical behaviour of the progress and confidence signals. It assesses whether the magnitudes and step-wise trends of $(\rho, c)$ are *internally consistent with the textual description of task progress and uncertainty*. High scores correspond to well-calibrated dynamics: $\rho$ increases as sub-tasks are resolved, $c$ increases when decisive evidence is observed and remains low for ambiguous sub-questions, and overall the numbers "make sense" as a faithful quantitative reflection of the explained state.

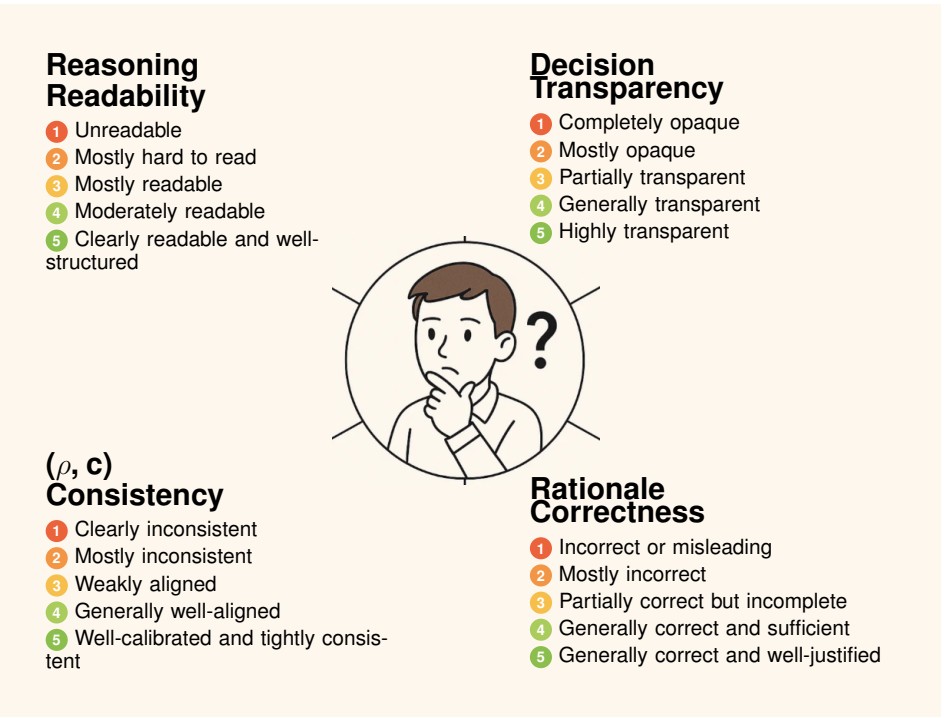

Figure 9: The four rubrics for evaluating ATDM's reasoning traces.

**Rationale Correctness** evaluates whether the reasoning at a given step is factually grounded in the underlying video and question, and *whether the cited evidence and causal explanation objectively justify the chosen action* ("answer" vs. "wait"). High scores require both factual accuracy (no hallucinated events or incorrect descriptions) and a logically appropriate use of evidence, such that the selected segments/events and the corresponding explanation are necessary and sufficient to support the decision.

Table 10: **Expert evaluation** of the reasoning process of ATDM on StreamingBench. The data in the table represents the *average score* for each indicator. The detailed indicators and the meaning of each corresponding score are in Table 12.

| Expert | Readable | Transparency | $\rho/c$ Matching Degree | Correctness |
|---|---|---|---|---|
| Qwen2.5-VL-72B | 4.6 | 4.2 | 4.3 | 4.0 |
| GPT-4o | 4.5 | 4.2 | 4.0 | 3.8 |
| Average | 4.55 | 4.20 | 4.15 | 3.90 |

### A.7 SUB-QUESTIONS RELEVANCE AND TYPE CORRECTNESS

**Expert Evaluation.** To further verify that ATDM generates task-relevant and structurally meaningful sub-questions, we conduct an additional expert evaluation using the LLM judge (GPT{4o). Given the original question and the corresponding list of ATDM-generated sub-questions, the LLM is prompted to assess each sub-question along two dimensions: 1) task relevance to the main question, and 2) the correctness of its semantic *type* label, which is required by our decomposition prompt of Part-2 in Section F. For each sub-question, the judge returns a scalar task-relatedness score in $\{1, \ldots, 5\}$, a boolean flag indicating whether the declared type is appropriate.

**Task relevance** is designed to quantify how strongly a sub-question contributes to solving the original main question, assuming that it can be answered correctly. We adopt a 1–5 Likert scale with the following rubric: 1-Completely unrelated, 2-Mostly unrelated, 3-Partially related, 4-Clearly related, 5-Strongly task-critical.

**Type correctness** evaluates whether the declared sub-question type matches the semantics of the sub-question content. The LLM judge is instructed to output a binary decision (correct / incorrect).

We further compare three prompt variants: (a) the full model with relevance/observability constraints, (b) a no-constraint variant that removes phrases enforcing explicit task relevance, and (c) a free-exploration variant that additionally encourages broad, unconstrained decomposition. As reported in Table 11, relaxing the constraints consistently degrades both LLM-judged task-relatedness and downstream streaming accuracy (e.g., from $71.60\%$ for the full model to $69.58\%$ and $68.40\%$ for the no-constraint and free-exploration prompts, respectively). These results are aligned with the failure modes discussed in our concurrent analysis (Jang et al., 2025): indiscriminate sub-questioning increases reasoning complexity without improving task utility, whereas ATDM's controlled Part-2 prompt reliably steers the model toward focused, task-critical sub-questions.

Table 11: Evaluation for relevance of sub-questions and the prompt's influence in Part-2.

| Avg Task relevance | Avg Type correctness | Full Prompt | w.o. Requirement | "Freely" |
|---|---|---|---|---|
| 4.971/5 | 0.9992/1 | 71.60 | 69.58 | 68.40 |

## B  MORE DISCUSSION AND FUTURE WORK.

**(1) Depth-as-memory.** We encourage treating Transformer *depth* as staged memory for streaming: assign progressively stronger aggregation to deeper segments and study depth-aware schedules with mask-guided fusion across backbones. This line of inquiry, consistent with hierarchical/aggregation evidence for long-range spatiotemporal reasoning and robustness, merits systematic, model-agnostic exploration. **(2) Decision timing and control.** We advocate explicit, calibrated internal signals in LLMs—e.g., progress $\rho$ and confidence $c$—to govern *when* to answer, wait, or reflect, moving beyond ad-hoc "longer CoT." This connects naturally to work on model self-calibration and self-reflection. **(3) Extensions from video QA.** (i) *Multi-stream evidence alignment:* equip each modality (vision, audio, motion, text) with its own HPSI-style memory and fuse per-stream $\rho/c$ via a lightweight controller to decide when joint evidence is sufficient—useful under missing frames and hard cuts. (ii) *Open benchmarks for decision quality:* complement accuracy/latency with explicit scoring of *decision timing* and *evidence alignment* (e.g., rewarding on-time, well-supported answers) for the video understanding task.

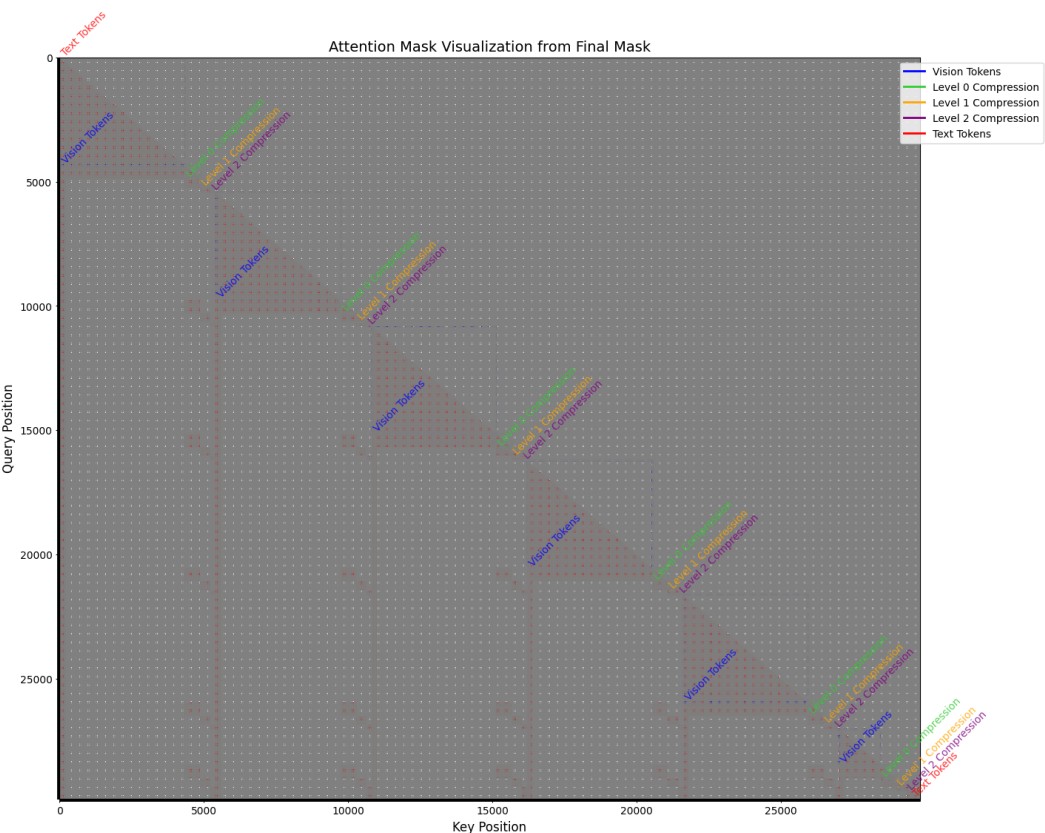

Figure 10: A real example of the attention mask in our final $1L/3$ layer of the LLM. The ratio between the original video tokens and the 3-level aggregated tokens is depicted. Compared to the original video input tokens, the proportion of aggregated tokens we introduce is minimal. As shown in this figure and Fig. 2, our custom attention mask guides the model in hierarchically allocating attention across different visual regions, fostering progressive focus on the visual tokens themselves for improved video understanding in LLMs.

Table 12: Evaluation rubric for ATDM's reasoning traces along four dimensions, each scored from 1 (worst) to 5 (best).

| Metric | Score | | | | |
|---|---|---|---|---|---|
| | **1** | **2** | **3** | **4** | **5** |
| **Reasoning Readability** | **Unreadable** | **Mostly hard to read** | **Moderately readable** | **Clearly readable** | **Highly readable** |
| | Severely disorganized or largely irrelevant to the requested content, with contradictory or unrelated statements forced into the same step; the overall meaning is very hard to recover. | Some segments are understandable, but the output is only weakly related to the requested content or to previous sub-questions; the context feels jumpy or redundant, and captions are not clearly written. | Each part broadly follows the instructions; there are no blatant off-topic or highly confusing jumps, and captions are mostly coherent, though some text shows weak contextual linkage. | Each part strictly follows the instructions; contextual relevance is clear, captions are coherent and sufficiently detailed, and answers align well with the caption. | All parts strictly follow the instructions with strong cross-part coherence, forming a globally consistent and logically connected reasoning process. |
| **Decision Transparency** | **Completely opaque** | **Mostly opaque** | **Partially transparent** | **Generally transparent** | **Highly transparent** |
| | Even after reading the full reasoning and inspecting the relationship between $(\rho, c)$ and sub-task states, the rater cannot see how they relate to main question nor why answers or waits at this step; decisive cues are missing or sub-tasks appear irrelevant. | The high-level relation to the main question is intelligible, but the updates of sub-tasks are unclear; if active-thinking steps are present, update reasons are opaque and the causal chain is confusing. | The rater can basically follow the overall reasoning, the sub-task updates, and the rough causes of $(\rho, c)$ changes; if reflection steps exist, the causal chain is largely reasonable, though some details remain implicit. | It is reasonably clear when evidence is considered sufficient or insufficient, and why $(\rho, c)$ and sub-task states are updated; if reflection steps exist, the causal chain before and after the update is clear and coherent. | The trace makes explicit *when and why* the decision is triggered, *which evidence* makes it sufficient to answer now, and *how* the current sub-task states and $(\rho, c)$ jointly justify answering versus waiting; the timing logic is easy to understand. |
| $(\rho/c)$ **Consistency** | **Clearly inconsistent** | **Mostly inconsistent** | **Weakly aligned** | **Generally well-aligned** | **Well-calibrated and tightly consistent** |
| | $(\rho, c)$ are often at odds with the reasoning (e.g., very high progress when the text emphasises strong uncertainty, or very low progress when most sub-tasks are stated as resolved); mismatches are frequent and severe. | Some steps look plausible, but overall $(\rho, c)$ rarely align with the described state; across the full trace, their magnitudes or changes feel arbitrary or untrustworthy. | The coarse trend of $(\rho, c)$ roughly follows the narrative (e.g., both increase as evidence accumulates), but many local steps feel off (e.g., unexplained spikes or plateaus). | For most steps, the level and evolution of $(\rho, c)$ are consistent with the textual description of progress and uncertainty; minor mismatches exist but do not substantially undermine trust in the signals. | Step-wise changes in $(\rho, c)$ closely track the reasoning: $\rho$ rises as sub-tasks are completed, $c$ rises when decisive evidence appears and stays low for unresolved or ambiguous sub-questions. |
| **Rationale Correctness** | **Incorrect or misleading** | **Mostly incorrect** | **Partially correct but incomplete** | **Generally correct and sufficient** | **Clearly correct and well-justified** |
| | The rationale contains clear factual errors, hallucinates events that do not occur, or relies on irrelevant or contradictory evidence; it fails to justify the chosen action and may even support the opposite. | Some parts touch on relevant content, but important evidence is missing or misinterpreted; the causal story is weak or flawed, so the decision is only very weakly supported. | The rationale captures several correct and pertinent aspects, but omits critical evidence or leaves key causal links under-specified; the decision is somewhat supported, yet the justification is not fully convincing. | The cited evidence is largely accurate and relevant, and the overall reasoning provides a plausible, reasonable explanation that covers most key factors needed to solve the task. | The rationale is factually accurate, targets the right segments/events, and the sub-question answers are correct while highlighting, from multiple angles, the key factors needed to solve the question. |

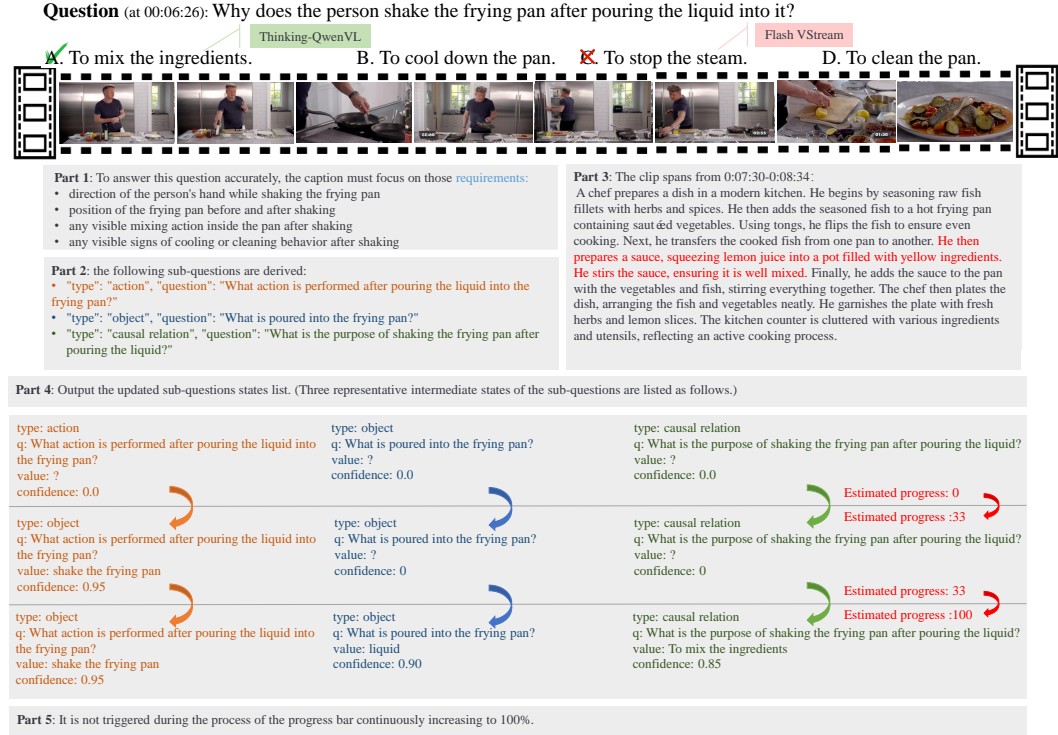

Figure 11: An example illustrating the outputs of each ATDM component in Thinking-QwenVL; in this case, the model's response confidence increases monotonically, so Part-5 (active thinking for reflection) is not triggered.

## C  ADDITIONAL VISUALIZATIONS

In addition to the examples presented in the main text, we provide further decision-making illustrations using ATDM for both Thinking-QwenVL and Flash-VStream in Fig. 11 & 12. We also include concise examples of cases that trigger *active thinking*, to clarify the outputs produced by each ATDM component and to demonstrate their specific roles across the two models.

## D  ETHICS STATEMENT

This work strictly adheres to the ICLR Code of Ethics. No human-subjects studies or animal experimentation were conducted. All datasets used for training and evaluation were sourced from the open-source community and used in compliance with their licenses and usage guidelines; no personally identifiable information was collected or processed. We took care to assess and mitigate potential biases and discriminatory outcomes, and we performed no experiments that could raise privacy or security concerns. We are committed to transparency and integrity throughout the research process.

## E  LLM USAGE

Large Language Models (LLMs) were used solely to assist with writing—primarily for grammar correction and minor phrasing edits to improve coherence and readability. The LLM did not participate in ideation, research methodology, experimental design, data analysis, or interpretation of results. All research concepts and analyses were conceived, executed, and validated by the authors. The authors take full responsibility for the content of the manuscript, including any text revised with LLM assistance. We verified that all LLM-assisted text complies with ethical guidelines and does not introduce plagiarism or scientific misconduct.

**Question** (at 00:01:10): What is the man doing while sitting on a chair?

A. Reading a book.   B. Polishing his shoes.   ✅ C. Adjusting his trousers and shoes.   D. Writing a letter.

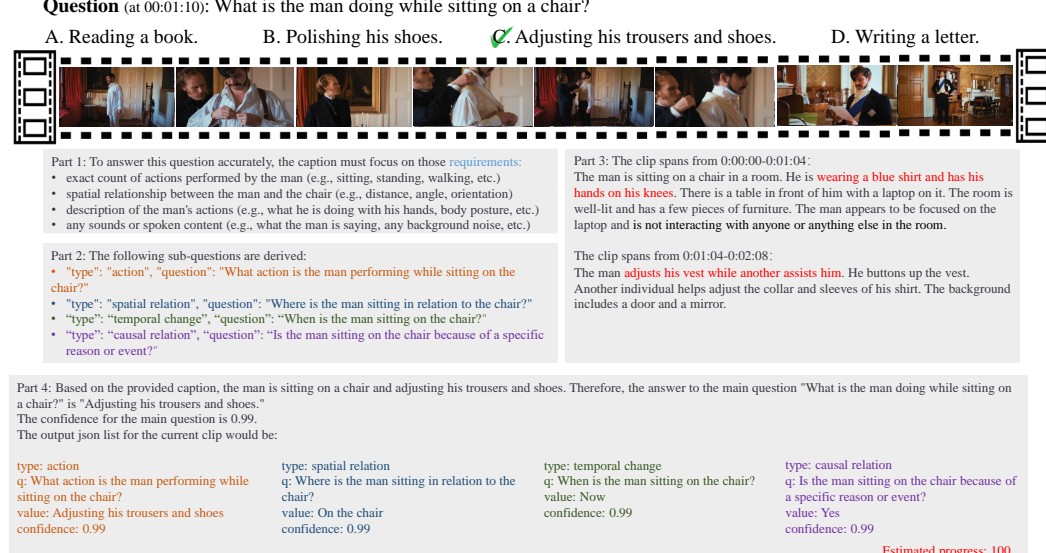

Part 1: To answer this question accurately, the caption must focus on those requirements:
- exact count of actions performed by the man (e.g., sitting, standing, walking, etc.)
- spatial relationship between the man and the chair (e.g., distance, angle, orientation)
- description of the man's actions (e.g., what he is doing with his hands, body posture, etc.)
- any sounds or spoken content (e.g., what the man is saying, any background noise, etc.)

Part 2: The following sub-questions are derived:
- "type": "action", "question": "What action is the man performing while sitting on a chair?"
- "type": "spatial relation", "question": "Where is the man sitting in relation to the chair?"
- "type": "temporal change", "question": "When is the man sitting on the chair?"
- "type": "causal relation", "question": "Is the man sitting on the chair because of a specific reason or event?"

Part 3: The clip spans from 0:00:00-0:01:04:
The man is sitting on a chair in a room. He is wearing a blue shirt and has his hands on his knees. There is a table in front of him with a laptop on it. The room is well-lit and has a few pieces of furniture. The man appears to be focused on the laptop and is not interacting with anyone or anything else in the room.

The clip spans from 0:01:04-0:02:08:
The man adjusts his vest while another assists him. He buttons up the vest. Another individual helps adjust the collar and sleeves of his shirt. The background includes a door and a mirror.

Part 4: Based on the provided caption, the man is sitting on a chair and adjusting his trousers and shoes. Therefore, the answer to the main question "What is the man doing while sitting on a chair?" is "Adjusting his trousers and shoes."
The confidence for the main question is 0.99.
The output json list for the current clip would be:

type: action
q: What action is the man performing while sitting on the chair?
value: Adjusting his trousers and shoes
confidence: 0.99

type: spatial relation
q: Where is the man sitting in relation to the chair?
value: On the chair
confidence: 0.99

type: temporal change
q: When is the man sitting on the chair?
value: Now
confidence: 0.99

type: causal relation
q: Is the man sitting on the chair because of a specific reason or event?
value: Yes
confidence: 0.99

Estimated progress: 100

Figure 12: An example illustrating the outputs of each ATDM component in Flash-VStream. The output of each part is influenced by the model's foundational visual comprehension and its ability to follow instructions.

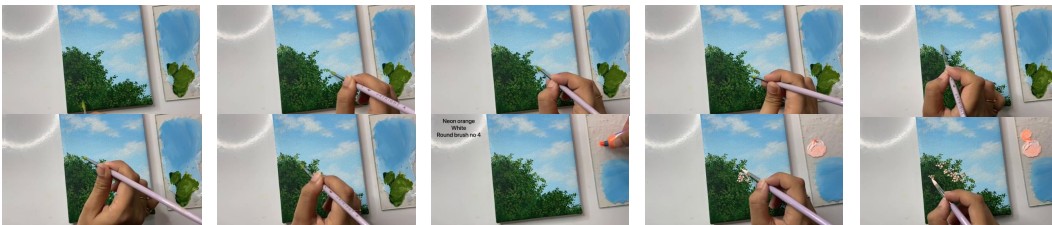

**Ours (with HPSI):**
A hand holding a round brush labeled 'Sap green Round brush no 4' applies green paint to a canvas. The canvas features a blue sky gradient with green foliage at the bottom. The hand **moves the brush from right to left**, adding details to the green area. A palette with blue paint is visible nearby. The hand **adjusts the brush's position and angle to enhance the foliage details**. The scene focuses on the meticulous application of paint to create a realistic tree outline against the blue sky background.

**Qwen2.5-VL:**
A hand holding a round brush no 4 applies sap green paint to a canvas, which already has a blue sky painted at the top. The artist carefully blends the green paint into the blue, creating a gradient effect. A palette with blue and green paint is visible nearby, indicating the mixing process. The hand moves methodically, adding texture and depth to the green area, suggesting an early stage in the landscape painting.

Figure 13: Comparison of model-generated captions for the same clip. Our caption explicitly encodes state changes over time ("moves from right to left", "adjusts position and angle"), which implies that the model is using historical visual memory and new frames to form a coherent, evolving narrative. The baseline, lacking hierarchical integration, mainly describes a single static scene.

# F   DETAILS OF THINKING-QWENVL'S PROMPT

Here, we provide detailed prompts of the five parts as well as their inputs and outputs. Question "*What is the width of the road right now?*" is as the example.

▶ **Part-1:**   The prompt for Part-1 *giving the instructions* for preparing for future steps should be:

---

### Part-1: Question-Guided Captioning Instructions

▶ **Input:**

<TASK DEFINITION>
Your task is to analyze the user's question and define EXACT observation requirements for video captioning from the video in order to help answer it.
Think carefully: What aspects of the video should a caption focus on to make answering this question possible?

<INSTRUCTIONS>
From the given question, generate a list of **observation requirements**: Each requirement should describe an important dimension that a future caption must pay attention to. Some CRITICAL FOCUS: 1. Quantification: Require exact counts when applicable 2. Directionality: Specify spatial relationships, positions and movement vectors 3. Object-anchored 4. Disambiguation of Confusable Concepts: If options include visually similar or easily confused concepts (e.g., "table" vs "counter", "cabinet" vs "shelf"), ensure captions distinguish them clearly through spatial context, object functions, or visual appearance.

For example: exact count of objects in someplace or the number of people, actions and their order, hand movements or object manipulation, specific visual details, interactions, gestures, spatial relationships, direction, distance, any sounds or spoken content
Such as: "exact count of apples placed in basket", "direction of sword thrust relative to opponent", "distance between white car and pedestrian when braking", "rotation angle of wrench during tightening"
<INSTRUCTIONS>

<CONSTRAINTS>
Only generate points that are visually observable. Do not speculate. Focus on fine-grained but relevant aspects. Max 5 points. Return your result in this **JSON** format:
{ "question": [What is the width of the road right now?], "caption requirements": [ "<|quantifiable requirement 1|>", "<|space observation requirement 2|>", "<|other observation point 3|>", ...,]}
<CONSTRAINTS>

<TASK DEFINITION>

▶ **Output:**

```
{
    caption_requirements: [
        width measurement of the road,
        position of the road within the frame,
        any obstacles or landmarks for scale reference,
        any changes to the road width over time
    ]
}
```

▶ **Part-2:**    The detailed prompt for Part-2 is shown as follows:

---

Part-2: Question Decomposition

▶ **Input:**

<TASK DEFINITION>
Your ONLY goal in this step is to read the user's main question below. Break it down into a set of precise, concrete sub-questions. Each sub-question should focus on a specific, observable aspect of the video (e.g., object, person, action, spatial relation, etc.). These sub-questions represent the key elements that must be visually or aurally verified in the video to answer the main question.

<CONSTRAINTS>

- Only include attributes that are **explicitly required** or clearly implied by the question.
- Do NOT use background knowledge, commonsense, or speculate.
- Do NOT include any explanations or commentary.
- Output must be in **valid JSON**, under the top-level key "required_attributes".
- **Do not include trailing commas.**

Return your result in this JSON format: {"question": [What is the width of the road right now?], "required subquestions": [{"type": <|type|>, "description": <|Required Subquestion description|>]}
<CONSTRAINTS>

<TASK DEFINITION>

▶ **Output:**

```
{
  required_subquestions: [
    { type: object,           question: Is there a road visible in
    the video? },
    { type: temporal_change,  question: Is the road width
    consistent throughout the video, or does it change over time? },
    { type: spatial_relation, question: Is the road width measured
    from edge to edge, or from center to center? },
    { type: other,            question: Is there any measurement
    tool used to measure the road width? }
  ]
}
```

---

In our sub-question decomposition prompt in Part-2, requirements such as "focus on a specific, observable aspect of the video" and "explicitly required or clearly implied by the question" are explicitly designed to avoid failure modes where the model generates freely diverging sub-questions with only weak relevance to the main query. Instead of allowing free-form decomposition, ATDM tightly controls both the *generation* and the *use* of sub-questions. The prompt guides the model to propose only a small set (typically 3–5) of sub-questions that are directly derived from the main question and cover it from complementary, semantically aligned perspectives. Beyond this constrained construction, ATDM incorporates two refinement mechanisms in Part 4 and Part 5: Part 4 continuously updates each sub-question's answer as new clips arrive, while Part 5 explicitly corrects them using cross-clip evidence. Together, these mechanisms ensure that both the generation and use of sub-questions remain well controlled.

▶ **Part-3:**   The detailed prompt for Part-3 is shown as follows. To convey the overall message, we present some content of clip captions before `0:07:46` here:

> A bustling city street is captured during a rainy day. **The road is wide**, with multiple lanes for traffic. Vehicles, including yellow taxis and various cars, navigate through the wet asphalt. ... A few pedestrians walk along the sidewalks, while vehicles move steadily despite the rain. The scene transitions from a more open area to a busier intersection with more traffic and pedestrians. **The road remains consistently wide throughout**, with clear lane markings and traffic flow.

---

**Part-3: Video Clip Captioning**

▶ **Input:**

<TASK DEFINITION>
Watch the current video clip and generate a descriptive caption, you must focus your caption on the following key observation points:
```
<|Caption Requirements List|>
<|Video|>
```

<INSTRUCTIONS>
1. Base your caption only on what is clearly visible. Limit to 500 words max.
2. Be specific and concrete: describe actions, hand use, counts, object states, etc.
3. Use short, factual sentences. No speculation, no vague summaries.
4. Precision first if required: `"three apples"` NOT `"some fruits"`, `"thrusting downward at 45° angle"` NOT `"attacking"`, `"2.3m left of tree"` NOT `"near tree"`, etc.

<CONSTRAINTS>
Return the caption in valid JSON format:

```
{
   clip_timestamp: {timestamp},
   caption: detailed caption that fulfills the requirements
}
```

<CONSTRAINTS>

▶ **Output:**

```
{
   clip_timestamp: 0:07:46-0:08:50,
   caption: A busy urban street scene during rainy weather. The road
     is wet and reflective, with multiple lanes of traffic moving in
      both directions. Skyscrapers line both sides of the street,
    creating a dense cityscape. Pedestrians cross the street at a
    crosswalk, some holding umbrellas. Vehicles, including yellow
    taxis and various cars, navigate through the traffic. The road
    surface shows signs of wear but remains functional. There are no
     visible construction or obstacles affecting the road width. The
      overall atmosphere reflects a bustling city environment despite
      the rain.
}
```

---

▶ **Part-4:**   The full prompt used in Part-4 is provided below. For brevity, we omit the complete problem statement and the intermediate outputs referenced in earlier parts.

**Part-4: Sub-answer Extraction and Filling Information**

▶ **Input:**

<TASK DEFINITION>
Your task is to:
1. Read the main user question and the list of required **subquestions** (from Part-1).
2. Read the caption of the current video clip.
3. For **each subquestion**, determine whether the caption provides enough information to answer it:
- If yes: provide an appropriate answer ('value') and a confidence score between 0 and 1.
- If no or uncertain: set "value": "?" and "confidence": 0.0.

<INPUT>
**Main Question:**
`<|Question|>`
**Required Subquestions** (from Part-2 or latest output from Part-4):
`<|Required Subquestions|>`
**Caption of the current clip:**
`<|Past caption|>`

<OUTPUT FORMAT>
Return one top-level JSON object with the key "`subquestion_status`".
Each item must include:
- "type": one of `["object", "attribute", "person", "action", "scene", "event", "temporal change", "spatial relation", "causal relation", "count", "other"]`
- "question": the original subquestion (from Part-1)
- "value": the answer extracted from the caption (or "?" if not found)
- "confidence": a float between 0 and 1

Also include an overall "`estimated_progress`" field (e.g., percentage of subquestions with confidence $\geq 0.85$).

<OUTPUT TEMPLATE>

```
{
  subquestion_status: [
    {
      type: <attribute_type>,
      question: <subquestion_text>,
      value: <answer_or_?>,
      confidence: 0.xx
    },
    ...
  ],
  estimated_progress: <int from 0 to 100>
}
```

<CONSTRAINTS>
- Only rely on what is explicitly visible or audible in the current caption.
- Do NOT use prior background knowledge or context.
- Do NOT speculate or fabricate.
- Ensure output is valid JSON (no trailing commas).
- If nothing is observed, return all values as "?" with "confidence": 0.0.

▶ **Output:**

---

**Part-4: Sub-answer Extraction and Filling Information**

(Here, we present only a single representative intermediate state.)

```
{
   subquestion_status: [
     {
       type: object,
       question: Is there a road visible in the video?,
       value: yes,
       confidence: 0.95
     },
     {
       type: temporal_change,
       question: Is the road width consistent throughout the video,
   or does it change over time?,
       value: consistent,
       confidence: 0.90
     },
     {
       type: spatial_relation,
       question: Is the road width measured from edge to edge, or
   from center to center?,
       value: ?,
       confidence: 0.0
     },
     {
       type: other,
       question: Is there any measurement tool used to measure the
   road width?,
       value: no,
       confidence: 0.85
     }
   ],
   estimated_progress: 75
}
```

Self-triggered reflection is a deterministic cross-clip reasoning step that revises answers, progress, and confidence when confidence-based triggers indicate high uncertainty or significant semantic shifts. In the prompt of Part-5, the "self-triggered reflection" stage is implemented as a cross-clip reasoning pass built on the confidence signals $c$. Given the decomposed sub-questions, their current answers, the corresponding $(\rho, c)$ values, and the recent relative clip captions, reflection (i) restructures these local pieces into a globally coherent reasoning chain across clips, and (ii) revises sub-question answers and $(\rho, c)$ whenever single-clip evidence is unreliable or mutually inconsistent. This prevents the controller from being myopically tied to a single clip and yields a smoother, more globally consistent reasoning trajectory.

Reflection is triggered purely by the quantitative signals, in particular by the confidence $c \in [0, 1]$ and its change $|\Delta c|$ between adjacent clips. The threshold $c \approx 0.5$ represents the boundary between "clearly relevant" and "not clearly relevant" visual evidence for a sub-question. Large $|\Delta c|$ indicates a major semantic shift between clips, while persistently low $c$ suggests that the required evidence is distributed across multiple clips rather than concentrated in the current or some single one. In both cases, we trigger reflection to explicitly re-examine and consolidate evidence across clips. For decision commitment, ATDM uses a higher threshold $c > 0.85$: this encodes that the model believes the currently accumulated evidence is both relevant and sufficient, while still leaving room for skepticism so that hypotheses formed from a single-clip perspective are not over-committed. Together with the progress signal $\rho$, these thresholds define a normalized $[0, 1]$ semantics for "relevance" and "sufficiency", and they directly govern when ATDM keeps waiting, when it invokes reflection, and when it finally decides that the visual evidence is *first sufficient*.

▶ **Part-5:** The detailed prompt for Part-5 is shown as follows. Then, we provide two specific examples of the output.

---

Part-5: Active Thinking for Refining the Reasoning across Clips

▶ **Input:**

<TASK DEFINITION>

**1. Cross-clip causal reasoning**
- Analyze each new clip caption for **direct** evidence related to each attribute.
- Build an explicit, ordered chain **only** for attributes with relevant evidence. Use arrow notation: "Clip X → [supports/contradicts/provides evidence for] [attribute] because [exact caption text]". If a clip provides no relevant evidence for any attribute, state: "Clip X → No relevant evidence for current attributes".

**2. Evidence relevance check**
- For each attribute, explicitly check whether the captions contain relevant information. Mark attributes as "relevant evidence found" or "no relevant evidence".

**3. Update the attribute list**
- **Preserve** original values and confidences for attributes without relevant evidence. Modify attributes **only** where direct, explicit evidence is found; quote the exact caption text that supports the change.

<INPUT>
**Question:**
`<|Question|>`
**Latest reasoning state** (attribute list + confidences):
`<|Past CoT State|>`
**Past clip captions when confidence changes** (chronological order):
`<|Clip Captions|>`

<OUTPUT FORMAT>
Return exactly **one** JSON object (no extra prose).

```
{
  causal_chain: [
    Clip X -> [reasoning or 'No relevant evidence']
  ],
  attribute_status: {
    <Attribute-Type>: {
      value: <value>,
      confidence: 0.xx,
      status: upgraded | downgraded | unchanged | newly_added,
      note: old 0.yy -> new 0.xx, [exact caption evidence] |
    unchanged - no evidence
    }
  },
  estimated_progress: <percentage with confidence > 0.5>
}
```

<CONSTRAINTS>
**Remember: NO EVIDENCE = NO CHANGE.**
- No evidence ⇒ status "unchanged", original confidence preserved.
- Quote the **exact** caption text in notes when updating.

▶ **Output:**

---

```
{
  causal_chain: [
    Clip 3 -> [provides evidence for] [Is there a black SUV directly
    in front of the camera right now? because A vehicle directly
    ahead is visible, appearing to be a dark-colored car.]
  ],
  attribute_status: {
    Is there a black SUV directly in front of the camera right now?:
    {
      value: A vehicle directly ahead is visible, appearing to be a
    dark-colored car.,
      confidence: 0.75,
      status: upgraded,
      note: old 0.6 -> new 0.75, [A vehicle directly ahead is visible
    , appearing to be a dark-colored car.]
    }
  },
  ...,
  estimated_progress: 50
}
```

```
{
  causal_chain: [
    Clip 4 -> [provides evidence for] [is there a painting visible on
     the wall? because] 'A close-up view of an artist's hand
    meticulously painting details on a canvas.',
    Clip 4 -> [provides evidence for] [is text readable on the
    painting? because] 'Text 'IT'S IN THE DETAILS' appears
    prominently over the artwork.',
    Clip 4 -> [provides evidence for] [is the camera focused on the
    lower left corner of the wall while showing the painting? because
    ] 'Text 'IT'S IN THE DETAILS' appears prominently over the
    artwork.'
  ],
  attribute_status: {
    is there a painting visible on the wall?: {
      value: yes,
      confidence: 0.95,
      status: upgraded,
      note: old 0.00 -> new 0.95, 'A close-up view of an artist's
    hand meticulously painting details on a canvas.'
    },
    is text readable on the painting?: {
      value: yes,
      confidence: 0.95,
      status: upgraded,
      note: old 0.00 -> new 0.95, 'Text 'IT'S IN THE DETAILS' appears
     prominently over the artwork.'
    },
    is the camera focused on the lower left corner of the wall while
    showing the painting?: {
      value: yes,
      confidence: 0.95,
      status: upgraded,
      note: old 0.00 -> new 0.95, 'Text 'IT'S IN THE DETAILS' appears
     prominently over the artwork.'
    },
  },
  estimated_progress: 95
}
```

