# OpenReview forum: "Progressive Online Video Understanding with Evidence-Aligned Timing and Transparent Decisions"
_ICLR.cc/2026/Conference — ICLR 2026 Poster_

### Official Review · Reviewer_EPyy · 2025-10-28

**Soundness:** 3
**Presentation:** 3
**Contribution:** 3
**Rating:** 8
**Confidence:** 4

**Summary:**

This paper introduces a novel framework for online video understanding in streaming scenarios, where visual agents must respond to queries precisely when sufficient evidence first emerges in a video stream. The proposed system, Thinking-QwenVL (built on Qwen2.5-VL-7B), decouples reasoning control from memory integration through two core modules: 1. Active Thinking Decision Maker (ATDM): A transparent reasoning controller that decomposes queries into sub-questions, tracks observable metrics like progress (ρ) and confidence (c), and self-triggers reflections for cross-clip causal updates. 2. Hierarchical Progressive Semantic Integration (HPSI): An efficient memory system using learnable multi-level aggregation tokens inserted at different decoder depths with structured sparse attention. Extensive experiments on online benchmarks (e.g., StreamingBench, OVOBench, OVBench, RTVBench) show significant improvements.

**Strengths:**

1. The paper effectively bridges the gap between offline and real-world streaming video understanding by emphasizing evidence-aligned timing and transparency.
2. HPSI's hierarchical aggregation reduces token overhead while preserving cross-clip relations and causal consistency, making it suitable for long videos under tight budgets.
3. Extensive evaluations across multiple benchmarks validate the approach, with clear improvements over state-of-the-art models.

**Weaknesses:**

1. Built on Qwen2.5-VL-7B, the results may not generalize to smaller or different architectures. What about the performance in smaller-sized models.
2. Focuses primarily on visual evidences potentially lacks diversity. Have you considered the multi-modal settings?

**Questions:**

What about the computational overhead of the multi-stage ATDM process?

---

> ### Author Response · Authors · 2025-11-26
> **Response to Reviewer EPyy (1/2) - W1&Q1**
>
> > **R4 – W1**: Built on Qwen2.5-VL-7B, the results may not generalize to smaller or different architectures. What about the performance in smaller-sized models.
>
> ### **Thinking-QwenVL with HPSI+ATDM also works well on a 3B backbone, and the design is architecture-agnostic rather than Qwen-specific**.
>
> Empirically, we show that **applying HPSI+ATDM to a smaller 3B backbone** yields strong performance that is competitive with or better than similarly sized models and **only moderately below 7B Dispider on StreamingBench**. Methodologically, both HPSI only require a standard multi-layer LLM with self-attention and ATDM only requires instruction-following capability, the two do not rely on any Qwen-specific component.
>
> ### 1. Our 3B model with HPSI+ATDM remains competitive with or better than similar-sized models across benchmarks.
>
> We apply the full HPSI+ATDM framework to a smaller backbone, **Qwen2.5-VL-3B (“Ours-3B”)**, and evaluate it on offline long-video and online benchmarks:
>
> | Model | VideoMME | MLVU | LongVideoBench | StreamingBench |
> | --- | --- | --- | --- | --- |
> | VILA1.5 3B | 42.2 | 44.4 | 42.9 | – |
> | LongVU 3.2B | 51.5 | 55.9 | – | – |
> | Phi-3.5-Vision-4.2B | 50.8 | – | – | – |
> | **Ours-3B (HPSI+ATDM)** | **51.44** | **56.2** | **51.01** | **62.62** |
>
> From this Table:
>
> - On **StreamingBench**, Ours-3B achieves **62.62%**, which is **only about 5 points below** the 7B Dispider model, while clearly outperforming prior 8B streaming baselines such as VideoLLM-Online (35.99%).
> - On **offline long-video benchmarks** (VideoMME, MLVU, LongVideoBench), Ours-3B is **competitive with or better than** similarly sized or even slightly larger models.
>
> These results indicate that **the benefits of HPSI+ATDM are not limited to a single 7B backbone**: the same design also yields strong performance in the 3B regime, suggesting good scalability to smaller models.
>
> ### 2. HPSI and ATDM only require a multi-layer LLM with attention and instruction following, and are not tied to Qwen-specific components.
>
> From a methodological standpoint, **our framework is designed to be architecture-agnostic**:
>
> - **HPSI** only assumes a **deep, multi-layer LLM with its attention**. As illustrated in Fig. 2, we:
>     - split the LLM layers into three depth segments (shallow / middle / deep), assign each segment a different aggregation strength (coarse → mid → strong), and
>     - use attention masks to guide progressive token aggregation across layers.
>
>     This mechanism does **not** depend on any Qwen-specific module.
>
> - **ATDM**, likewise, is a **prompt-level decision controller** that only requires a basic video-understanding capability, and a reasonable level of **instruction-following**.
>
>     We also apply ATDM to **non-Qwen** video-LLMs (e.g., LLaVA-based Flash-VStream in Table 5), and it consistently improves their streaming performance.
>
> Together, these points support that **both HPSI and ATDM are conceptually portable**: they exploit generic properties of LLM architectures and instruction-following VLMs, rather than any Qwen2.5-VL–specific internals.
>
> ### 3. We have added the 3B results on StreamingBench in Table 5 to the revised manuscript to make this generalization explicit.
>
> - Added the **Ours-3B** row (Qwen2.5-VL-3B with HPSI+ATDM) to the StreamingBench comparison table, and
> - Highlighted in the text that **our framework maintains strong performance on a 3B backbone on p.10**, reinforcing that the method is not restricted to large Qwen2.5-VL-7B models.
>
> We hope this clarifies that our approach generalizes well to smaller models and is not tied to a single architecture size or vendor-specific backbone.
>
> ---
> ---
> >R4 –Q1: What about the computational overhead of the multi-stage ATDM process?
>
> ### **The inference cost of  ATDM requires no concern given the model's high token throughput**.
>
> We present a complete cost breakdown of the 5 ATDM parts (average over videos) on an Nvidia A100 GPU:
>
> | Part in ATDM | P-1 | P-2 | P-3 | P-4 | P-5 | **Total** | **FPS** | **Tokens/s** | **Latency (s)** | **Acc (%)** |
> | --- | --- | --- | --- | --- | --- | --- | --- | --- | --- | --- |
> | **Inference cost (avg tokens / video, ours)** | 51.40 | 87.25 | 1051.87 | 549.40 | 264.50 | **2004.42** | **8.49** | **1261** | **13.2** | **71.60** |
> | Flash-VStream | – | – | – | – | – | – | 8.45 | 16 | 9.5 | 22.53 |
>
> Most computational cost is concentrated in Part 3/4; other stages are lightweight. And the wrapper-style design and high throughput make the multi-stage process feasible for our task. There is only **13.2s total latency per video at 8.49 FPS**, **3.7s more** than the simpler Flash-VStream pipeline at similar FPS (8.45 / 8.49). In return, model’s accuracy on StreamingBench increases  from **22.53% to 71.60%**.
>
> We have also added **Figure 8** and an **efficiency-comparison paragraph on p.10** in the revised manuscript.

---

> ### Author Response · Authors · 2025-11-26
> **Response to Reviewer EPyy (2/2) - W2**
>
> >**R4 – W2**: Focuses primarily on visual evidences potentially lacks diversity. Have you considered the multi-modal settings?
>
> ### **ATDM/HPSI are designed to be modality-agnostic; we intentionally evaluate in a vision-only setting for fairness, and multimodal extension is a natural next step.**
>
> ### 1. Our current experiments are intentionally vision-only to ensure fair comparison with video-centric baselines.
>
> Thank you for raising this point. In the current paper, we **deliberately restrict all experiments to visual streams only**, mainly to ensure **fair and apples-to-apples comparison** with existing streaming baselines such as Flash-VStream, VideoLLM-Online, and Dispider, which are all evaluated in **video-centric, vision-only** settings. Under this protocol, we can cleanly attribute gains to **our decision controller (ATDM)** and **hierarchical aggregation (HPSI)**, without conflating them with additional audio or textual cues.
>
> ### 2. ATDM’s evidence-timed decision process naturally accommodates multimodal evidence.
>
> Conceptually, ATDM operates on **abstract “evidence” and quantitative state** rather than exclusively on visual tokens. The controller:
>
> - decomposes the task into sub-questions,
> - accumulates progress ρ and confidence c for each sub-task, and
> - decides **when** evidence is sufficient to answer.
>
> This logic does **not** require the evidence to be visual. In a multimodal setting, the same ATDM loop can be applied with:
>
> - **audio-aware / motion-aware sub-questions** in Part 2 (e.g., “What sound is heard when …?”, “How does the motion pattern change?”) alongside visual ones, and
> - **additional non-visual features** (audio embeddings, motion descriptors, or background text snippets) injected as **extra evidence sources** in Parts 4–5 when visual confidence is low or fluctuating.
>
> In other words, ATDM already provides a **modality-agnostic, evidence-timed decision interface**; only the **evidence channels** (visual vs. audio vs. text) change.
>
> ### 3. HPSI’s progressive token aggregation readily extends to multi-stream multimodal memory.
>
> Similarly, HPSI only assumes a **deep LLM with attention over sequences of tokens**. The aggregation mechanism assigns different aggregation strengths at each level and progressively compresses raw tokens into a compact, semantically rich memory. It is **agnostic to the token modality**. In a multimodal extension, one can:
>
> - maintain **separate HPSI memories per modality** (e.g., video, audio, sensor signals), and
> - build a **fused multimodal memory** that ATDM reads from when computing ρ and c.
>
> Reflection (Part 5) then becomes a **cross-modal coordination step**, reconciling evidence from different streams (e.g., when audio and video disagree) without changing the overall controller design.
>
> ### 4. We have added an explicit multimodal discussion in a more future-work paragraph in the revised paper on p.18.
>
> We hope this clarifies that our framework is **not limited to visual evidence**,  with multimodal extensions being a natural and important direction we are actively considering.

---

> > ### Comment · Reviewer_EPyy · 2025-11-28
> > **Response to authors' rebuttal**
> >
> > Appreciate for the authors' rebuttal.
> >
> > The rebuttal has solved most of my concerns, and the experimental results seem convincing to me. I will keep my initial score.

---

### Official Review · Reviewer_arDz · 2025-10-30

**Soundness:** 3
**Presentation:** 3
**Contribution:** 3
**Rating:** 4
**Confidence:** 3

**Summary:**

The paper introduces Thinking-QwenVL, a framework for streaming video understanding. It consists of two core components: 1) Active Thinking Decision Marker (ATDM) : a module that determines when to provide an answer based on task progress and confidence. 2) Hierarchical Progressive Semantic Integration (HPSI): an efficient memory system for streaming videos that uses multi-level learnable aggregation tokens to capture video content effectively. Thinking-QwenVL demonstrates strong performance across various online and offline video understanding benchmarks.

**Strengths:**

- The paper is well written and of high quality.

- The proposed method is both powerful and elegant. The hierarchical processing of visual signals allows progressive handling of long video features from coarse to fine levels. Moreover, decomposing questions into sub-questions and leveraging confidence scores enhances flexibility while preserving the base model’s capabilities.

- The experiments are extensive and comprehensive; including both online and offline benchmarks effectively demonstrates the framework’s robustness and versatility.

**Weaknesses:**

Despite its effectiveness, several concerns and missing details remain:

- In HPSI, it is unclear how the authors decided on the number of aggregation levels (i.e., three). Does this configuration balance efficiency and accuracy? From Table 4, the performance gain from levels 2–3 appears marginal, as the third row (only level 1) performs comparably to the full method. Further clarification would help justify the design choice.

- In ATDM, the use of sub-questions and confidence scores is crucial, yet the paper lacks sufficient detail to understand how the module operates. Prior work [1] shows that sub-questioning can increase complexity or even degrade performance if applied indiscriminately. Have authors faced such issues? Furthermore, how were the thresholds (e.g., 0.85 and 0.5) for confidence scores in Parts 4 and 5 determined? A clearer interpretation is needed.

- In Table 1, Thinking-QwenVL performs worse than its backbone model (Qwen2.5-VL), but this issue is not addressed, which contradicts the claim in L410–414.

- The recent SOTA method [2] is missing from the comparison, and Thinking-QwenVL shows inferior performance to [2].

*I would consider revising the rating if I misunderstood any part, and the authors clarify these issues in the rebuttal phase.*

**References**

[1] Confidence-guided Refinement Reasoning for Zero-shot Question Answering, arXiv 2025

[2] StreamBridge: Turning Your Offline Video Large Language Model into a Proactive Streaming Assistant, arXiv 2025

**Questions:**

See the Weaknesses.

---

> ### Author Response · Authors · 2025-11-26
> **Response to Reviewer arDz (1/4) - W1**
>
> > **R3 – W1**: In HPSI, it is unclear how the authors decided on the number of aggregation levels (i.e., three). Does this configuration balance efficiency and accuracy? From Table 4, the performance gain from levels 2–3 appears marginal, as the third row (only level 1) performs comparably to the full method. Further clarification would help justify the design choice.
>
> ### **Three aggregation levels are necessary to outperform single-step pooling under a fixed token budget and to balance token cost and accuracy**.
>
> **Under the *same final token budget* and overall aggregation strength**, a three-level HPSI hierarchy clearly outperforms single-step pooling (+7.4% absolute), and that the seemingly **strong “only level 1” baseline actually keeps 3× more final tokens, thus trading efficiency for accuracy**. In other words, levels 2–3 are precisely what allow us to perform strong compression without suffering the larger accuracy drop seen in the single-step pooling baseline.
>
> ### 1. Under a fixed token budget, three-level HPSI significantly outperforms single-step pooling.
>
> It is important to distinguish between **how aggressively we compress tokens** and **how many tokens remain at the end in Table 4**:
>
> - **Full 3-level HPSI (Row 1).**
>
>     Aggregation pattern: `initial → 3 → 2 → 1`.
>
>     This performs **strong, progressive aggregation** and yields a **small final token budget** (≈0.1× of the original tokens under 93.75% aggregation strength).
>
> - **Single-step pooling (Row 5).**
>
>     Aggregation pattern: `initial → 1`.
>
>     Here we enforce **the same overall aggregation strength** as the full model but apply it in **one shot**, with the **same final token budget** (also ≈0.1× of the original).
>
> Under this matched-budget setting, the accuracy difference is:
>
> | Token-aggregation method | Aggregation pattern | Relative final token budget | Row in Table 4 | Acc (%) |
> | --- | --- | --- | --- | --- |
> | **Full HPSI (3 levels)** | `→3 →2 →1` | **1×** | 1 | **56.3 (+7.4)** |
> | Single-step pooling | `→1` | **1×** | 5 | 48.9 |
>
> Thus, **three-level HPSI improves accuracy by +7.4 points over single-step pooling under the same compression ratio and token budget**, showing that a multi-stage hierarchy is substantially more effective than compressing everything in a single step. (The straightforward one-step pooling aggregation method serves as the baseline for other variant aggregation approaches.)
>
> ### 2. “Only level 1” appears close in accuracy because it uses 3× more tokens and therefore much weaker aggregation.
>
> With our attention mask, this variant effectively behaves as:
>
> - Aggregation pattern: approximately `initial → 3 → 3 → 3`, i.e., **no progressive strengthening** of aggregation;
> - It therefore keeps **3× more final tokens** than the full 3-level model and corresponds to **much weaker overall aggregation**.
>
> Summarized together:
>
> | Token-aggregation method | Aggregation pattern | Relative aggregation strength | Relative final token budget | Row in Table 4 | Acc (%) |
> | --- | --- | --- | --- | --- | --- |
> | **Full HPSI (3 levels)** | `→3 →2 →1` | strong, progressive | **1×** | 1 | **56.3** |
> | “Only level 1” + our mask | `→3 →3 →3` | weak, non-progressive | **3×** | 3 | 54.7 |
>
> The fact that “only level 1” is close to full HPSI **under a much larger token budget** shows that three-level HPSI achieves almost the same accuracy while using only about one third of the final tokens. In other words, **levels 2–3 are precisely what allow us to perform strong compression without suffering the larger accuracy drop seen in the single-step pooling baseline.**
>
> ### 3. Three aggregation levels form the minimal coarse–mid–fine hierarchy that matches LLM depth and yields a good efficiency–accuracy trade-off.
>
> Beyond raw numbers, our design goal for HPSI is to exploit the **vertical depth of the LLM** by assigning **different aggregation roles to different layers**, instead of forcing all aggregation into a single step: Shallow layers → Middle layers → Deep layers.
> The three levels are therefore **not three independent tricks**, but **three stages of a single, progressive process** that gradually moves from dense visual tokens to a very small, semantically rich set **without discarding long-range information**. A single-step pooling layer cannot disentangle these roles and is forced to compress everything at once, which empirically leads to the larger accuracy drop observed in the single-step baseline.
>
> ### 4. We have added a conceptual paragraph in Sec. 3.1 on p.5 to clarify this design rationale of the revised manuscript.
>
> We hope this clarifies why we adopt the design in HPSI and how it balances efficiency and accuracy under realistic token budgets.

---

> ### Author Response · Authors · 2025-11-26
> **Response to Reviewer arDz (2/4) - W2**
>
> >**R3 – W2**: Prior work [1] shows that sub-questioning can increase complexity or even degrade performance if applied indiscriminately. Have authors faced such issues? Furthermore, how were the thresholds (e.g., 0.85 and 0.5) for c determined?
>
> ### **ATDM uses controlled, refined sub-questioning with interpretable confidence thresholds, avoiding the failure modes reported in [1]**.
>
> ATDM’s sub-questioning is strictly **controlled**—we use a task-relevant set of sub-questions with built-in verification and update—so we **do not observe the degradation reported in [1]**. On the contrary, across backbones such as Qwen2.5-VL and Flash-VStream, sub-questioning consistently improves streaming performance in Table 1 and 5. The thresholds 0.5 and 0.85 are chosen as interpretable “relevance” and “safe-to-commit” regions in a normalized [0,1] space and are kept fixed across backbones without brittle sensitivity.
>
> ### 1. Uncontrolled, irrelevant sub-questions without refinement cause failures in [1], not sub-questioning per se.
>
> As you note, [1] shows that **indiscriminate sub-questioning can hurt performance**. Their negative findings occur under two explicit conditions:
>
> - Sub-questions are often **irrelevant to the main question** or drift away from what is actually needed to answer it.
> - There is **no mechanism to verify or refine** these sub-QAs, so inaccurate or noisy sub-answers.
>
> In their own words (p.1 in [1]):
>
> > “Our investigation reveals that **when subQAs are irrelevant to the main question** or paired with inaccurate answers, they introduce noise into the reasoning process, ultimately degrading answer quality. Despite these risks, prior works (….) **often incorporate sub-QAs without sufficient verification or refinement**, as their relevance to the main question is neither assessed nor guaranteed.”
> >
>
> Importantly, **[1] does not conclude that “sub-questions are inherently harmful”**; after diagnosing these failure modes, they themselves propose a more careful sub-questioning strategy. The message is that **sub-questions must be constructed and refined carefully**, not that sub-questioning should be abandoned.
>
> **This is exactly the perspective we adopt in ATDM.**
>
> ### 2. ATDM uses the task-relevant, continuously updated, and refined collection of sub-QAs, so we do not observe the degradation reported in [1].
>
> - **We constrain sub-questions to be directly task-related from some observable aspects**.
>
>     In Part 2 (decomposition), the model is guided to produce sub-questions under the prompt, which explicitly asks to “focus on specific, observable aspects” and “only include attributes that are explicitly required or clearly implied by the question”.
>
> - **We add two built-in refinement mechanisms (Parts 4 & 5) which act as verification and refinement layers** that [1] explicitly identifies as missing.
>
>     - **Part 4** continuously updates each sub-question’s status when more evidence becomes available.
>     - **Part 5 (reflection)** explicitly revisits and corrects earlier answers using cross-clip evidence.
>
> ### 3. Confidence thresholds 0.5 and 0.85 encode interpretable “relevance” and “safety” regions and work robustly across backbones.
>
> These values are therefore **not arbitrary magic numbers** but encode a **task-level interpretation** of “relevance” (≈0.5) and “sufficiency/safety” (>0.85) in a normalized [0,1] space. In our experiments, we keep these thresholds **fixed across different backbones** and do not observe brittle sensitivity around them.
>
> - **(c ≈ 0.5): relevance boundary.**
>
>     The values near **0.5** are the **boundary between “clearly relevant” and “not clearly relevant”** visual evidence for a given sub-question. When c hovers in this region or remains low over several clips, ATDM interprets this as “the current clip is not clearly informative for this sub-question,” which is precisely the regime where reflection (Part 5) is considered to aggregate evidence across clips instead of over-committing to a local view.
>
> - **c > 0.85: safe-to-commit region.**
>
>     High confidence indicates that the model believes the available evidence is both relevant and sufficient. In our streaming setting, each Part-4 update only sees a **local, partial view** (a single new clip plus accumulated state), so we intentionally choose a **conservative “safe” region** c > 0.85:
>
>     - this requires strong belief in the current evidence before committing,
>     - yet still leaves room for later correction via reflection if future clips contradict the current hypothesis.
>
>     Together with the progress indicator ρ, this threshold controls **when ATDM stops waiting and outputs a final answer** for each sub-question and ultimately for the main question.
>
> ### 4. We have added paragraphs on p.24 for part 2 and p.27 for part 5  in the revised manuscript to clarify these design choices and threshold interpretations.

---

> ### Author Response · Authors · 2025-11-26
> **Response to Reviewer arDz (3/4) - W3**
>
> > **R3 - W3**: In Table 1, Thinking-QwenVL performs worse than its backbone model (Qwen2.5-VL), but this issue is not addressed, which contradicts the claim in L410–414.
> >
>
> ### **R3 – W3: Thinking-QwenVL improves over Qwen2.5-VL under the same offline protocol (77.44% vs 73.68%); the 71.60% in Table 1 is from a stricter online streaming regime**.
>
> The apparent drop in Table 1 arises because **Qwen2.5-VL and Thinking-QwenVL are evaluated under different StreamingBench protocols**. Under the **same offline evaluation protocol**, our model **improves** over its Qwen2.5-VL backbone (77.44% vs 73.68%). The **71.60%** number in Table 1 comes from a **harder online/streaming setting**, so it does **not** indicate that Thinking-QwenVL harms the backbone.
>
> ### 1. Table 1 mixes offline backbone and online streaming models, so the scores are not directly comparable.
>
> In Table 1, StreamingBench contains two different regimes:
>
> - **Qwen2.5-VL (backbone row) is evaluated under a standard offline long-video protocol**:
>
>     the model observes the current whole raw video (or dense key frames) and only answers **after** seeing all evidence.
>
> - **Thinking-QwenVL (our row)** and other streaming baselines such as Flash-VStream / Dispider **are evaluated under an online streaming protocol**:
>
>     the model must **answer along the stream**, operate under a **memory budget**, and rely on its **internal memory** rather than re-processing all raw frames.
>
>
> By construction of StreamingBench, the **online/streaming regime is strictly harder than the offline regime**: the model faces partial observability and timing constraints, whereas the offline backbone does not. Therefore, it is expected that **online streaming accuracy (71.60%) is lower than offline accuracy (73.68% and 77.44%)**, and this gap reflects **task difficulty**, not a degradation of the underlying backbone.
>
> ### 2. Under the same offline protocol, Thinking-QwenVL is strictly better than the Qwen2.5-VL backbone.
>
> To make the comparison fair, we also run our model under the **same offline protocol** as the Qwen2.5-VL backbone on StreamingBench:
>
> | Model | Protocol | Acc (%) |
> | --- | --- | --- |
> | Qwen2.5-VL | offline | 73.68 |
> | Ours | offline | **77.44 (+3.76)** |
> | Ours | online | 71.60 |
>
> Under **identical offline conditions**, our method **improves** the backbone from **73.68% → 77.44%**.
>
> The **71.60%** reported in Table 1 is the accuracy under the more demanding **online streaming protocol**, where we obtain SOTA performance among streaming models (e.g., Dispider 67.63% → ours 71.60%). Thus, the claim in L410–414 is specifically about **online/streaming SOTA**, and there is **no contradiction** once the evaluation regimes are distinguished.
>
> We greatly appreciate that you point out that there is a possibility of misunderstanding the original statement around L410–414, so we have revised the sentence in the “Streamingbench” paragraph:
>
> - **Original** (**L409–410**; by default, comparisons are against models in the same streaming category):
>
>     > “Compared to previous models, we have improved the state-of-the-art performance by 3.97%, increasing the accuracy from 67.63% to 71.60%.”
>     >
> - **Revised (L417–419, more precise):**
>
>     > “Compared to previous online and streaming video models, we have improved the state-of-the-art performance by 3.97% **on the online task**, increasing the accuracy from Dispider’s 67.63% to our 71.60%.”
>     >

---

> ### Author Response · Authors · 2025-11-26
> **Response to Reviewer arDz (4/4) - W4**
>
> >**R3 – W4:** The recent SOTA method [2] is missing from the comparison, and Thinking-QwenVL shows inferior performance to [2].
>
> ### **StreamBridge [2] is not included in Table 1 due to different [multi-turn vs. single-turn] processing streaming videos experimental settings.**
>
> We fully acknowledge StreamBridge [2] as a strong and very relevant concurrent framework for streaming video understanding. However, the **experimental setting of StreamBridge is fundamentally different from ours and from the baselines we compare against**, so a direct accuracy comparison would be misleading rather than informative.  Our task is to decide first-sufficient visual evidence time and then answer immediately, so we require single-pass video processing and multi-round processing is not suitable for this task configuration.
>
> ### 1. Different evaluation protocol: multi-turn vs. single-turn streaming video input settings.
>
> As described in their Section 3 of StreamBridge, their framework turns offline Video-LLMs into streaming assistants that **process videos at 1 FPS in a *multi-turn* manner**, maintaining historical context within a token budget and answering across multiple rounds of interaction.   In their own words in P7 of their paper:
>
> > “In contrast, with the StreamBridge framework, …,  these offline models are equipped to process streaming videos at 1 FPS in a multi-turn manner, while maintaining input length and historical contexts within a predefined maximum token budget.”
> >
>
> In contrast:
>
> - Our main comparisons (Flash-VStream, VideoLLM-Online, Dispider, and ours) follow a **single-pass, single-turn** streaming video protocol on StreamingBench:
>     - The model **passes through the video once**,
>     - Must **respond at the appropriate moment** in that single pass,
>     - And cannot “revisit” previous frames except via its compressed memory.
> - The task of our model is decide first-sufficient visual evidence time and then answer immediately. This matches the task definition of typical **streaming / online** settings used by Flash-VStream and Dispider, where the emphasis is on **one-pass evidence alignment and memory aggregation**, not on multi-turn processing over the same content.
>
> Because StreamBridge uses a different **multi-turn interaction protocol and task setting**, where the model can repeatedly process and reason about the same underlying content over several turns, its accuracy is **not directly comparable** to our single-turn setting. Mixing these numbers into one table would give the impression that all models were evaluated under identical constraints, which is not the case.
>
> ### 2. We restrict Table 1 to single-pass StreamingBench baselines to preserve fairness and clarity.
>
> For fairness and clarity, we chose to:
>
> - Compare only against methods that share the **same single-pass, single-turn streaming protocol** (Flash-VStream, VideoLLM-Online, Dispider),
> - Use StreamingBench’s **one-shot response** setting, where each model answers *once per video* and must rely on internal memory rather than re-processing raw frames.
>
> Given these differences, we felt that listing StreamBridge side-by-side in our main quantitative tables—without fully re-running *all* baselines under their multi-turn setting—would be unfair to either side and could confuse the reader about what exactly is being measured and how the video is passed.
>
> We have **added the texts in ”Related Work“ section** to emphasize that our experimental setup (**single-turn**) differs from StreamBridge.

---

> > ### Comment · Reviewer_arDz · 2025-11-27
> > **Official Comments by Reviewer arDz**
> >
> > I appreciate the authors' effort to address my concerns and have further questions.
> >
> > - I appreciate the provided details of HPSI. I would suggest that the authors add a column to represent the relative final token budget for better understanding in Table 4.
> > - I didn't mean that ATDM wrongly constructs sub-QAs. It was about other findings while the authors were developing ATDM for effective sub-QAs construction, as I acknowledge its crucial role and effectiveness from Figure 7. However, I'm not fully convinced that the generated sub-QAs truly are task-related by such prompting, “focus on specific, observable aspects". Overall performance improvements are insufficient to yield the authors' claims.
> > - Also, I'm not convinced by the authors' response to the thresholds. So, how do the 0.5 and 0.85 values serve as a safe signal? **There is no quantitative analysis or evidence to prove this yet.** Furthermore, what is clear evidence for mentioning "do not observe brittle sensitivity around them" in the answer? Some empirical justification or sensitivity analysis are much needed.

---

> > > ### Author Response · Authors · 2025-12-02
> > > **Response to Reviewer arDz**
> > >
> > > Thank you very much for your careful follow-up questions and for your **positive feedback** on our clarifications for **Weakness 1,3,4**.  For your follow-up queries, we (i) add a column in Table 4 as you suggested, (ii) provide 2 empirical analyses on the task-relevance of sub-questions, and (iii) present quantitative sensitivity studies of the thresholds.
> > >
> > > ### 1. New LLM evaluation and 2 prompt ablations  directly evaluate and confirm the task-relevance of controlled sub-questions is highly.
> > >
> > > To directly address the core concern, whether our prompting scheme generates *truly task-related* sub-questions, we designed two new studies:
> > > 1. Direct Evaluation of Task-Relevance.
> > > 2. Ablations Linking Relevance to Performance.
> > >
> > > As a brief overview, we summarize in the table below, provide strong empirical support for our claims.
> > >
> > > | Avg task-relevance/ 5 | Avg Type correctness/1 | Full Model/100% | Prompt/w.o. requirement/100% | Prompt/added “freely”/100% |
> > > | --- | --- | --- | --- | --- |
> > > | 4.971 | 0.9992 | 71.60% | 69.58% | 68.40% |
> > >
> > > ### 1.1 New LLM Evaluation Confirms High Task-Relatedness of Sub-Question.
> > >
> > > For each question, the judge was tasked to:
> > >
> > > - Assign a task-relatedness score (1-5) for each sub-question (1=Completely Unrelated, 5=Strongly Task-Critical).
> > > - Assess type correctness, verifying if the sub-question's content matches its declared semantic type (0/1).
> > >
> > >
> > > Our empirical evidence further provides concrete, quantitative evidence that our sub-questions are highly task-aligned and structurally sound, validating our core design principle of generating a "small, directly task-relevant sub-question set.” The strong instruction-following of the baseline model serves as the foundation.
> > >
> > > ### 1.2 Prompt Ablations Prove Our Prompt Constraints are helpful.
> > >
> > > To directly test the reviewer's concern and connect with [1], we performed a critical ablation: what happens when we remove the constraints designed to ensure task-relatedness?
> > >
> > > We compared three prompting regimes:
> > >
> > > - Full Model (71.60): Our original Part-2 prompt with relevance constraints.
> > >
> > > - No-Constraint Prompt (69.58): Remove all above mentioned task-focused phrasing.
> > >
> > > - Free-Exploration Prompt (68.4): Constraints are removed and encourage to questioning unconstrainedly (e.g., “feel free to freely and fully explore your ideas and perspectives on deconstructing problems”,  mimicking the problematic setup in [1]).
> > >
> > > From the experimental results, it can be seen that as long as there is no explicit requirement for divergent output of sub-questions (including the number and content), our instructions can generate high-quality sub-questions that meet the task requirements on our model.
> > >
> > > We have added sentences in Sec. 3.2 (and discussion on p.19) to highlight it, and we reference [1] to emphasize our design is *aligned* with their recommendation for controlled, verified sub-questioning.
> > >
> > > ### 2. New cross-backbone and sensitivity analyses show that the 0.5 and 0.85 thresholds are robust, interpretable choices.
> > >
> > > We appreciate your insistence on empirical justification for the thresholds 0.5 and 0.85. Our intention was never to argue that these values are uniquely optimal, but rather that they are (i) *semantically meaningful* in the normalized ([0,1]) confidence space, and (ii) *robust* enough to be reused across backbones without brittle sensitivity.
> > >
> > > ### 2.1 A single pair of thresholds yields consistent gains across diverse streaming backbones.
> > >
> > > | Backbone | Flash-VStream | VideoLLM-Online | Ours-3B | Ours-7B |
> > > | --- | --- | --- | --- | --- |
> > > | Acc without ATDM | 22.53 | 35.99 | – | – |
> > > | Acc with ATDM | 26.58 | 41.50 | 62.62 | 71.60 |
> > >
> > > These results indicate that the chosen thresholds are *not* fragile hyperparameters tuned to a single architecture: a single ATDM configuration yields consistent improvements across heterogeneous backbones.
> > >
> > > ### 2.2 Threshold sensitivity analyses show a wide stable region without brittle peaks.
> > >
> > > Second, we conduct explicit sensitivity analyses over the thresholds used in Part 4 (update) and Part 5 (reflection). We vary one threshold at a time while keeping the rest fixed.
> > >
> > > | c in Part-5 | 0.7 | 0.6 | **0.5 (setting)** | 0.4 | 0.3 |
> > > | --- | --- | --- | --- | --- | --- |
> > > | Acc  | 70.40 | 72.02 | 71.60 | 71.49 | 70.14 |
> > >
> > > For Part 4 (update), we vary the “safe-to-commit” threshold over:
> > >
> > > | c in Part-4 | 0.95 | **0.85 (setting)** | 0.75 | 0.65 | 0.55 |
> > > | --- | --- | --- | --- | --- | --- |
> > > | Acc  | 71.80 | 71.60 | 71.44 | 70.94 | 70.30 |
> > >
> > > Taken together, these results clarify what we meant by “we do not observe brittle sensitivity”:
> > > - there exists a reasonably wide region in which performance remains high and stable; and
> > > - mis-setting the thresholds within this region does *not* cause catastrophic degradation.
> > >
> > > Our goal is therefore *not* to find “magic numbers”, but to treat progress and confidence as explicit, interpretable signals which can be reused across architectures and remain robust under reasonable perturbations.

---

### Official Review · Reviewer_iwc4 · 2025-10-31

**Soundness:** 3
**Presentation:** 3
**Contribution:** 3
**Rating:** 6
**Confidence:** 4

**Summary:**

This paper addresses the problem of online video understanding, focusing on two critical but often overlooked aspects: evidence-aligned response timing and decision transparency. The authors argue that conventional video LLMs, typically evaluated in offline settings, are ill-suited for real-world streaming scenarios where an agent must respond precisely when sufficient evidence becomes available. To tackle this, they propose a novel framework, Thinking-QwenVL, which decouples reasoning control from memory integration. The framework has two core components: The Active Thinking Decision Maker (ATDM): a reasoning controller that externalizes its decision process using explicit progress ($p$) and confidence ($c$) metrics, allowing it to align its response time ($t_r$) with the first-sufficient-evidence timestamp ($t^*$). The Hierarchical Progressive Semantic Integration (HPSI) module: an efficient memory system that uses multi-level aggregation tokens to maintain a global cognitive state under tight computational budgets. The paper demonstrates strong empirical results, achieving a new state-of-the-art on the StreamingBench benchmark.

**Strengths:**

- The paper clearly articulates and tackles the critical, practical challenges of response timing and decision transparency in streaming video analysis, which is a major step towards real-world applications.

- The ATDM module provides an effective framework for making the model's decision-making process transparent and quantifiable through progress ($p$) and confidence ($c$) scores. This is a significant strength for user trust and controllability.

- The HPSI module is a technically sound and well-executed solution for maintaining long-term, causally consistent context within a limited token budget, as demonstrated by its strong performance on long-video benchmarks.

- The paper achieves state-of-the-art results on several challenging online video benchmarks, most notably StreamingBench, providing strong evidence for the effectiveness of the proposed Thinking-QwenVL framework. The ablation studies are thorough and convincing.

**Weaknesses:**

- The primary weakness is the lack of analysis on the computational latency introduced by the ATDM module. The five-part reasoning process (generating instructions, decomposing the question, captioning, extracting answers, and reflecting) seems to require multiple LLM inference steps for each incoming video clip. This could create a significant processing bottleneck in a true real-time scenario, a concern that is not adequately addressed in the paper.

- The 5-part ATDM process, while transparent, appears highly structured and heavily engineered. This raises questions about its robustness and generalizability. How dependent is this structure on the specific base model (Qwen2.5-VL) and extensive prompt engineering? It's unclear if this complex chain-of-thought would transfer effectively to other video LLMs without significant re-tuning.

- The concept of "Active, Self-triggered Thinking" (Part 5 of ATDM) is an interesting idea but is not described in sufficient detail. The paper mentions it is triggered by low confidence or major semantic shifts, but the exact trigger conditions (e.g., thresholds, detection mechanisms) and the concrete steps of the "reflection" process are not clearly defined. This makes the mechanism less reproducible and its contribution harder to assess.

**Questions:**

- Could you provide an analysis of the wall-clock latency or computational overhead (e.g., number of forward passes per second of video) introduced by the ATDM module? How does this compare to simpler streaming models like Flash-VStream or Dispider, and how might it impact real-time performance?

- Can you elaborate on the development process for the 5-part ATDM prompt structure? How sensitive is the model's performance to the specific wording and structure of these prompts? Have you experimented with applying ATDM to other base models to test its generalizability?

- Could you provide a more detailed explanation of the "self-triggered reflection" mechanism? What are the specific criteria for triggering it (e.g., confidence thresholds, how are "major semantic shifts" detected)? What does the model do during reflection (e.g., does it re-process past clips, revise the question decomposition)?

---

> ### Author Response · Authors · 2025-11-26
> **Response to Reviewer iwc4 (1/3) - Q1&W1**
>
> > **R2- Q1 and W1** – The five-part reasoning process seems to require multiple LLM inference steps for each incoming video clip. Could you provide an analysis of the wall-clock latency or computational overhead introduced by the ATDM module?
>
> ### **ATDM adds only modest total latency (~3.7 s) while maintaining Flash-VStream-level FPS through pipelined execution of a few lightweight components. The per-clip overhead is dominated by one captioning pass plus one update pass, rather than five sequential LLM calls for every clip**.
>
> In our current implementation, ATDM adds about 3.7 seconds of total latency compared to Flash-VStream **on StreamingBench (A100)**, while **maintaining essentially the same frame processing speed (8.49 vs 8.45 FPS)**. This overhead arises from a small number of additional, pipelined reasoning steps, and we view it as a practical trade-off given the substantial accuracy gain (22.53% → 71.60%) and the explicit, controllable decision timing that ATDM provides.
>
> ### 1. Only 2 parts ---Part 3 (captioning) and Part 4 (sub-QA updating) are executed per clip.
>
> The five logical parts of ATDM are **not** all executed on every incoming clip:
>
> - **Part 3 (captioning)** and **Part 4 (sub-QA  updating)** are the only components that run **for every clip**.
> - **Part 1 (caption instruction)** and **Part 2 (question decomposition)** are one-off setup steps. The Part 3 and 4 continuously complete subtasks based on the objectives derived from 1 and 2 as the clip updates.
> - **Part 5 (self-triggered reflection)** is invoked **only when needed**, e.g., when confidence drops or there is a major semantic shift, so it is relatively infrequent in practice.
>
> ### 2. Pipelined execution keeps wall-clock latency low
>
> Implementation-wise, ATDM is organized as several **pipelined wrappers** instead of a single long sequential chain. On an A100 GPU:
>
> - Each wrapper uses **< 20 GB** of VRAM during inference, so we can run **multiple wrappers concurrently**.
> - In our deployment, we run:
>     - one wrapper continuously captioning the next clip,
>     - one wrapper updating sub-question answers and progress/confidence,
>     - and one wrapper on standby for occasional reflection.
>
> This design overlaps captioning and decision updates across clips, so although the logical reasoning has multiple parts, the **wall-clock latency per video** remains close to that of a simpler single-pass model.
>
> ### 3. Empirical latency and throughput comparison on StreamingBench provides experimental evidence for the feasibility of the wrapper-plus-parallel-pool method.
>
> We explicitly benchmarked our model against **Flash-VStream** on **StreamingBench** using an **NVIDIA A100** GPU. (Dispider’s full streaming pipeline is unfortunately not open-sourced, so an end-to-end latency comparison with Dispider is not currently possible.)
>
> | Model | Decision-making? | Transparent? | Controllable? | Decision method | Avg. total latency (s) | FPS | Throughput (tokens/s) | Acc. (%) |
> | --- | --- | --- | --- | --- | --- | --- | --- | --- |
> | Flash-VStream | No | – | – | – | 9.5 | 8.45 | 16 | 22.53 |
> | Dispider | Yes | No | No | Binary black-box gate | – | – | – | 67.63 |
> | **Ours** | Yes | Yes | Yes | ATDM | **13.2** | **8.49** | 1261 | **71.60** |
>
> **From this,**
>
> - **FPS:** Our model achieves **8.49 FPS**, essentially identical to Flash-VStream’s **8.45 FPS**, indicating similar frame processing capacity.
> - **Latency:** The **average total latency** increases from **9.5 s → 13.2 s**, i.e., an additional **≈ 3.7 s** over the simple model.
> - **Accuracy:** In return for this modest overhead, accuracy improves from **22.53% → 71.60%**, and the decision process becomes **explicit, interpretable, and controllable** via progress/confidence signals and optional reflection.
>
> In other words, ATDM turns a black-box, memoryless streaming pipeline into a transparent controller with only a small additional end-to-end latency, while preserving real-time frame throughput. Regarding Dispider, because its full binary-decision pipeline is not open-sourced, we cannot report a fair end-to-end latency number.
>
> ### 4. We have added Figure 8 and an efficiency-comparison paragraph on p.10 in the revised manuscript.
>
> - Added **Figure 8** and the **efficiency-comparison paragraph on p.10** that summarizes the latency/FPS/throughput vs. accuracy trade-off between Ours and Flash-VStream.
> - Modify and bold the sentence “Only Parts 3 and 4 require reasoning to be processed iteratively across video clips.” in Sec. 3.2 to emphasize that only Parts 3–4 are executed on every clip and that ATDM is implemented as a pipelined controller rather than five fully sequential passes per clip.
>
> Thank you for your question, we hope this clarifies that ATDM introduces only modest additional latency over simple streaming models.

---

> ### Author Response · Authors · 2025-11-26
> **Response to Reviewer iwc4 (2/3) - Q2&W2**
>
> >**R2- Q2 and W2**: How dependent is this structure on the specific base model (Qwen2.5-VL) and extensive prompt engineering? What is the development process for the 5-part ATDM prompt structure? How sensitive is the model's performance to the specific wording and structure of these prompts? How about the ATDM’s generalizability when applying it to other base models?
>
> ### **ATDM generalizes across 3 video-LLM frameworks and is designed as a portable, non-fragile decision controller rather than a Qwen2.5-VL-specific prompt hack**.
>
> ### 1. ATDM consistently improves decision-making across three heterogeneous video-LLM backbones.
>
> ATDM is **not** tied to Qwen2.5-VL or a particular backbone. When we attach the same ATDM controller to **three different video-LLM frameworks**—Flash-VStream (LLaVA), VideoLLM-Online (SigLIP + Llama-3-8B-Instruct), and our Qwen2.5-VL-3B/7B models—it consistently activates decision-making and improves online streaming performance:
>
> | Framework | Base Model | Acc (%) w/ ATDM |
> | --- | --- | --- |
> | Flash-VStream-7B | LLaVA | 26.58 (**+4.05**) |
> | VideoLLM-Online-8B | SigLIP + Llama-3-8B-Inst. | 41.50 (**+5.50**) |
> | Ours-3B | Qwen2.5-VL-3B | 62.62 |
> | Ours-7B | Qwen2.5-VL-7B | 71.60 |
>
> Both Flash-VStream and VideoLLM-Online are **non-Qwen** architectures with different vision encoders and decoding stacks, yet attaching ATDM still yields clear gains. **This indicates that ATDM behaves as a backbone-agnostic decision controller rather than a controller over-fitted to one specific model or codebase.** In our experiments, we observed that the only factor affecting ATDM reasoning texts arises in strict **JSON-format adherence: when we apply it to** Flash-VStream (the LLaVA baseline),  the model sometimes appends extra descriptive text beyond the requested JSON block, which mainly increases token cost rather than breaking the controller. Even in this case, as we show in the paper (**Fig. 12**), the outputs still capture key cues well enough for ATDM to exploit, and **adding our decision capability still brings noticeable gains** on StreamingBench.  These observations suggest that **ATDM is not fragile to the particular base model architecture**; instead, it benefits from any backbone that can reliably follow structured instructions and output a simple JSON schema.
>
> ### 2. The 5-part ATDM structure is a task-driven design for transparent, progress-aware decision control rather than ad-hoc prompt engineering.
>
> We emphasize that the 5-part ATDM process is a **task-driven** design for transparent, progress-aware decision control in online video understanding, rather than the result of ad-hoc prompt tinkering. Existing approaches such as Dispider expose only a binary 0/1 “answer vs wait” gate and thus offer no visibility into intermediate reasoning, while standard chain-of-thought, although interpretable, lacks explicit progress control and internal self-awareness of **how far** the model has advanced or whether the available evidence is sufficient, so decision-relevant information may fail to propagate along the stream.
>
> To address this, ATDM is organized around two explicit quantitative signals:
>
> - a **progress score** ρ indicating how far each sub-task has advanced, and
> - a **confidence score** c indicating how safe it is to commit.
>
> These variables compactly accumulate decision state, are propagated and continuously updated as new clips arrive, and operate on a decomposition of the original question into sub-questions that endows the task with an inherent notion of progress. In practice, we keep the prompt for **each part high-level and role-based**.
>
> The five parts thus form a **semantically modular pipeline**:
>
> 1. **Part 1** (instruction generation) uses **model-derived guidance** prompts to steer captioning toward truly important visual content, instead of relying on static hand-crafted instructions.
> 2. **Part 2** (question decomposition) turns the global query into sub-questions so that **progress can be explicitly tracked** per sub-task.
> 3. **Part 3** (captioning) summarizes each new clip to **provide fresh, localized evidence**.
> 4. **Part 4** (sub-QA update) maintains answers together with (ρ,c), turning an unstructured chain-of-thought **into a compact quantitative state that governs “answer vs wait”**.
> 5. **Part 5** (reflection) re-assesses past decisions under low confidence or major semantic shifts, **further stabilizing the reasoning process**.
>
> Empirically, as long as these **role breakdowns are preserved and the corresponding tasks are completed**, performance is robust across different frameworks. We therefore view the 5-part process as a **generic, task-aligned design for streaming decision control**, rather than a brittle, hand-crafted prompt script.
>
> ---
>
> ### 3. We have added Table 5 and a ‘robustness and applicability’ paragraph on p.10 in the revised manuscript to make ATDM’s generalizability explicit.

---

> ### Author Response · Authors · 2025-11-26
> **Response to Reviewer iwc4 (3/3) - Q3&W3**
>
> >**R2 – W3 and Q3**: Could you provide a more detailed explanation of the "self-triggered reflection" mechanism? What are the specific criteria for triggering it (e.g., confidence thresholds, how are "major semantic shifts" detected)? What does the model do during reflection (e.g., does it re-process past clips, revise the question decomposition)?
>
> ### **Self-triggered reflection is a deterministic cross-clip reasoning step that revises answers, progress, and confidence when simple confidence-based triggers indicate uncertainty or semantic shifts**.
>
> In our implementation, reflection is a deterministic cross-clip reasoning pass that takes the current decision state as input, re-examines evidence across recent clips, and revises answers, progress (ρ), and confidence (c) when simple thresholds on confidence magnitude and change indicate uncertainty or a major semantic shift.  Below we clarify (1) what reflection does, (2) how it is triggered, and (3) why it is reproducible and easily extensible.
>
> ### 1. Self-triggered reflection performs a cross-clip reasoning pass that can revise answers of the sub-questions with their (ρ,c).
>
> In our implementation, the key to reflection lies in **constructing chains of reasoning across clips**. It operates purely on the accumulated sub-questions’s state and recent clip summaries, **not on raw video**:
>
> - **Inputs.** Reflection receives as input:  the decomposed sub-questions and their current answers with (ρ,c); the captions of the clips (at which their c is related to triggering the reflection).
> - **Operation.** Given this state, reflection:
>     1. builds a coherent reasoning chain across these clips, explicitly aggregating evidence that may be distributed over time; and
>     2. revises sub-question answers and their (ρ, c) when associated evidence emerges.
> - **Outputs.** The updated answers and (ρ, c).
>
> This cross-clip pass prevents the controller from being myopically tied to any single clip and makes the reasoning trajectory smoother and more globally consistent over the video stream. We provide the exact Part-5 prompt and concrete reflection examples in the appendix, so the behavior of this pass is fully reproducible and easy to port.
>
> ---
>
> ### 2. Two quantitative confidence-based rules (confidence and its change) determine when its own reflection is activated.
>
> Reflection is triggered by two **quantitative confidence-based rules**:
>
> 1. **Persistently low confidence.** When c stays below 0.5 (indicating the single clip is insufficient for answering), reflection aggregates evidence across clips to revise or delay the decision.
> 2. **Large confidence changes. T**his change signals a **major semantic shift (visual scene transitions)**; reflection then integrates the new context with earlier evidence to correct or strengthen previous conclusions.
>
> We use the **same fixed thresholds** for c in all our experiments (including the above 3 different VLLM frameworks); given these thresholds and the provided Part-5 prompt, the reflection mechanism is deterministic and easy to replicate.
>
> ---
>
> ### 3. Reflection is reproducible from fixed prompts and thresholds and can be extended without altering the task structure.
>
> All five ATDM steps, including reflection, are documented in the appendix with **full prompts and example outputs**, so that other researchers can directly reproduce or migrate them to different video LLMs with minimal I/O changes. In our current design:
>
> - Reflection **does not change the sub-question decomposition** produced in Part 2, so the overall task structure remains stable.
> - Reflection **only updates** sub-question answers and their (ρ,c) scores, allowing the model to “change its mind” according to the new evidence chain while keeping the progress-tracking scaffold (Sub-Qs) intact.
>
> This separation keeps the mechanism modular and easy to reason about. At the same time, ATDM is flexible: by slightly adjusting the Part-5 prompt, one can extend reflection to revise sub-questions or perform additional reasoning operations if desired.
>
> Thank you for your suggestions. In the revised manuscript, we have **consolidated these details into two paragraphs before the prompt definitions on p.27**, and we explicitly describe the trigger rules and inputs/outputs of reflection to make its contribution easier to assess and reproduce.

---

### Official Review · Reviewer_5jps · 2025-11-01

**Soundness:** 3
**Presentation:** 3
**Contribution:** 3
**Rating:** 6
**Confidence:** 4

**Summary:**

This paper introduces Thinking-QwenVL, a new framework for online, evidence-aligned video understanding, addressing the key challenge of determining when a model should respond based on visual evidence as it streams. The framework explicitly separates reasoning control from memory integration, leading to two key components: Active Thinking Decision Maker (ATDM) and Hierarchical Progressive Semantic Integration (HPSI). The authors evaluate Thinking-QwenVL across several online and offline benchmarks (StreamingBench, OVOBench, RTVBench, OVBench, MLVU, VideoMME) and report significant improvements, e.g., +3.97% over Dispider on StreamingBench and +4.9% on OVOBench.

**Strengths:**

1. The work tackles a largely under-explored but crucial problem — real-time, evidence-aligned video understanding — distinct from traditional offline long-video reasoning.
2. The idea of decoupling reasoning control from memory integration is conceptually strong and practically motivated. The multi-level aggregation mechanism (HPSI) is an elegant solution for progressive semantic integration, offering a new perspective beyond standard pooling or RAG-based methods.
3. The paper is clearly written and well-structured, with informative visual diagrams (e.g., Fig. 1–4) illustrating timing, aggregation, and decision flow. Each module (ATDM, HPSI) is described step-by-step, making a complex architecture accessible.

**Weaknesses:**

1. While empirically strong, the paper could benefit from more conceptual analysis of why ATDM’s quantitative progress and confidence signals yield better evidence alignment. For instance, is the model implicitly learning uncertainty calibration or temporal gating?
2. The paper claims transparency and interpretability, but does not include user studies or objective interpretability metrics to support these claims (e.g., human evaluation of decision clarity or correctness of rationales).
3. Although ATDM aims for real-time decision-making, the paper does not explicitly report latency, throughput, or computational overhead compared to simpler baselines. Such results would help assess deployment feasibility for real-world streaming agents.
4. The ablations treat ATDM and HPSI largely separately, but it would be informative to analyze their synergy — for example, whether ATDM decisions remain robust if memory integration is partially disabled or simplified.

**Questions:**

1. For the reference in Latex, please use \citep instead of \cite to fix the reference issue in the main paper so that "Xun et al. (2025)" could be "(Xun et al. 2025)"
2. Have you conducted human or expert evaluations on whether ATDM’s progress/confidence signals improve user trust or understanding compared to black-box baselines?
3. What is the per-frame or per-second latency of Thinking-QwenVL relative to Dispider and Flash-VStream? How does hierarchical aggregation affect token throughput?
4. How does the model behave when the input stream contains missing frames, abrupt scene transitions, or noisy temporal cues? Does ATDM maintain stability in such conditions?

---

> ### Author Response · Authors · 2025-11-26
> **Response to Reviewer 5jps (1/3) - W1, Q1, Q2&W2**
>
> >**R1 – W 1**: While empirically strong, the paper could benefit from more conceptual analysis of why ATDM’s quantitative progress and confidence signals yield better evidence alignment.
>
> **ATDM converts past reasoning into a compact, history-aware state that enables robust temporal gating beyond a memoryless 0/1 classifier.**
>
> **Conceptually**, ATDM augments the model with a *quantitative state* of its ongoing reasoning process. Instead of making an independent 0/1 decision for each new visual chunk, the model maintains, for every sub-question, two continuous signals — a progress score ρ (“how far this sub-task has advanced”) and a confidence score c (“how safe it is to commit now”). **These signals summarize the history of past decisions and evidence into a low-dimensional state that is far richer than a single binary gate, and they are fed back into subsequent steps so that the model can revise or refine earlier decisions as new evidence arrives.** In this way, ATDM turns an unstructured chain-of-thought into an explicit, compact representation of task progress that directly guides when to answer versus when to wait in streaming video.
>
> In terms of **implementation**, at each ATDM step the model outputs, for each sub-question, its current answer together with (ρ, c). The full set of sub-question answers and their (ρ, c) values from the previous step is then concatenated with the latest clip caption and fed as input for the next step (see the prompt format in the appendix).  Compared to Dispider-style purely visual binary classifiers that only see the current clip, ATDM therefore operates on a **history-aware quantitative state** that accumulates and stabilizes over time, which we find crucial for aligning decisions with the *first sufficient* visual evidence in online settings.
>
> The **ablation experiment** P-4 in Figure 7 isolates this effect. In P-4 we remove the explicit progress and confidence and **replace each decision step with a prompt-guided binary decision**, while keeping the rest of the system intact. This seemingly small change leads to **a 3.62% performance drop**, indicating that continuous progress/confidence signals provide more informative and stable decision guidance than ad-hoc 0/1 decisions.
>
> Regarding uncertainty calibration and temporal gating, we intentionally use c as an approximate uncertainty indicator and the joint pair (ρ, c) as a lightweight temporal gate that decides “answer now vs wait.” ATDM is not designed as a fully calibrated probabilistic model or theoretically optimal stopping rule; instead, it is a deliberate design choice that forces the model to explicitly represent how far it believes it has progressed and how confident it is, and to condition future actions on these quantities.
>
> In the **revised paper**, we **add a paragraph in Sec. 3.2** that makes this conceptual role explicit and clarifies how ATDM provides a practical, history-aware mechanism for temporal gating in online video understanding (new text marked in blue).
>
> >**R1-Q1**:  We have completed the replacement (’\citep’) as per your recommendation. Thank you so much for the reminder.
>
> > **R1 – Q2 and W2** : Expert evaluations of ATDM’s decision clarity and rationales
>
> To systematically assess whether ATDM genuinely improves users' understanding and trust in the model’s behaviour, we use **four complementary metrics** to focus on both **decision clarity** and **rationale correctness.  We provide the full metrics in Figure 9, Table 11 and the Sec A.6 in the appendix.**
>
> Specifically, we decompose these aspects into **four metrics** applied to ATDM’s generated reasoning traces:
>
> - **(A) Decision Clarity**
>     1. **Readability**: whether a user can understand *why* the model decides to answer or continue waiting at a given step from the full reasoning texts.
>     2. **Transparency**: whether the explicit progress/confidence scores (\rho, c) together with the intermediate rationales make the process more understandable.
>     3. **(\rho/c) Matching Degree**: whether the numerical values of (\rho) and (c) are perceived as consistent with the current task state and the textual rationale.
> - **(B) Rationale Correctness**
>
>     4. **Rationale Correctness**: whether the current reasoning chain clearly and correctly supports the decision taken at that step (i.e., evidence and causal explanation are appropriate for “answer” vs. “wait”).
>
> All four metrics are rated on a **1–5 Likert scale** (1 = poorest, 5 = most complete).  We use **LLM judges** (GPT-4o and Qwen2.5-VL-72B).
>
> | Expert | Readable | Transparency | ρ /c Matching degree | Correctness |
> | --- | --- | --- | --- | --- |
> | Qwen2.5-VL-72B | 4.6 | 4.2 | 4.3 | 4.0 |
> | GPT-4o | 4.5 | 4.2 | 4.0 | 3.8 |
> | Avg | 4.55 | 4.20 | 4.15 | 3.90 |
>
> We greatly appreciate your suggestions. We have incorporated the specific definitions of the evaluation metrics and the corresponding results into **Table 10, Table 11 and Figure 9**.

---

> > ### Author Response · Authors · 2025-11-26
> > **Response to Reviewer 5jps (2/3) - W3&Q3**
> >
> > >**R1 – W3 and Q3**: Although ATDM aims for real-time decision-making, the paper does not explicitly report latency, throughput, or computational overhead compared to simpler baselines. Such results would help assess deployment feasibility for real-world streaming agents.
> >
> > ### **Comparable FPS to Flash-VStream with a Tunable Speed–Accuracy Trade-off**
> >
> > On an A100 GPU, Thinking-QwenVL with ATDM+HPSI **matches Flash-VStream’s average FPS** (8.49 vs 8.45 frames/s) while incurring only ~3.7 s additional end-to-end decision latency (13.2 vs 9.5 s), and hierarchical aggregation exposes a smooth speed–accuracy trade-off by controlling frame aggregation and token throughput.
> >
> > ### 1. **Per-frame / per-second latency is comparable to the Flash-VStream model.**
> >
> > On StreamingBench with an NVIDIA A100, our default configuration (93.75% aggregation) achieves **8.49 frames/s**, while Flash-VStream reaches **8.45 frames/s**. In terms of end-to-end decision latency (time from the beginning of the stream to the final answer), our model averages **13.2 s**, compared to **9.5 s** for Flash-VStream. Thus, the explicit reasoning and decision-control pipeline in ATDM+HPSI introduces only about **+3.7 s** of latency while operating at essentially the **same throughput** as Flash-VStream. We believe this overhead is well-justified, since in the same setting our method reaches **71.6%** accuracy on StreamingBench, whereas Flash-VStream reaches only **22.53%**. (Dispider’s full streaming pipeline has not yet been open-sourced.)
> >
> >   ### 2.  As the **aggregation rate increases** from 75% to 97.5%, **token throughput** increases from 786 to 1351 tokens/s and **FPS** increases from 5.28 to 9.12 frames/s.
> >
> > To isolate the impact of HPSI, we vary the **hierarchical aggregation rate** and report both **average FPS** and **average token throughput** (tokens/s):
> >
> > | Method | Aggregation (%) | Avg FPS (frames/s) | Avg Throughput (tokens/s ×100) |
> > | --- | --- | --- | --- |
> > | Ours | 97.50 | 9.12 | 13.51 |
> > | Ours | 96.88 | 9.09 | 13.47 |
> > | Ours | 95.83 | 8.77 | 13.00 |
> > | **Ours (paper setting)** | **93.75** | **8.49** | **12.61** |
> > | Ours | 87.50 | 7.24 | 10.77 |
> > | Ours | 83.33 | 6.63 | 9.92 |
> > | Ours | 75.00 | 5.28 | 7.86 |
> > | Flash-VStream | N/A | 8.45 | 0.16 |
> > | Ours-3B (small model) | 93.75 | 7.07 | 10.50 |
> >
> > As the **aggregation rate increases** from 75% to 97.5%, **FPS** increases from 5.28 to 9.12 frames/s and **token throughput** increases from 786 to 1351. This provides a **continuous control knob**: users can adopt a slightly lower aggregation rate (e.g., 93.75%, our default) for a stronger accuracy–latency balance, or push aggregation higher when they prioritize speed and compute efficiency. The 3B variant (“Ours-3B”) at the same 93.75% aggregation still achieves **7.07 frames/s** and **1050 tokens/s**, showing that ATDM+HPSI also remains deployable under tighter compute budgets.
> >
> > Overall, these measurements indicate that **Thinking-QwenVL is practically real-time** under the StreamingBench protocol: it matches Flash-VStream’s FPS on A100, adds only modest extra latency, and in return delivers substantially stronger evidence-aligned reasoning with transparent decision control.
> >
> > We have **visualized this table as** **Figure 8** and added a short **efficiency-comparison paragraph on p.10** of the revised paper to make these trade-offs clearer (new text marked in blue).

---

> > > ### Author Response · Authors · 2025-11-26
> > > **Response to Reviewer 5jps (3/3) - W4&Q4**
> > >
> > > > **R1 – W4 and Q4**: How does the model behave when the input stream contains missing frames, abrupt scene transitions, or noisy temporal cues, or memory integration is partially disabled or simplified? Does ATDM maintain stability in such conditions?
> > >
> > > ### **ATDM–HPSI exhibit synergistic yet non-fragile behavior and remain stable under missing frames and simplified memory integration.**
> > >
> > > We appreciate the reviewer’s request for a deeper analysis of robustness and synergy. Empirically, we find that (i) **ATDM’s decisions remain stable** when the input stream contains missing frames and temporally noisy cues, and (ii) the ATDM–**HPSI combination yields consistent gains but is not brittle**: even when we simplify the memory hierarchy, accuracy degrades only mildly rather than collapsing. (iii) **HPSI can encode state changes over time and thereby enables ATDM** to leverage its own temporal history to make evidence‑aligned decisions rather than reacting myopically to each local glitch.
> > >
> > > ### 1. Our model remains robust under missing frames and simplified HPSI from the experiment’s perspective.
> > >
> > > We conduct a unified set of experiments on **StreamingBench**:
> > >
> > > - **Missing / noisy temporal cues.**
> > >
> > >     After extracting videos at 1 FPS, we uniformly subsample the stream to keep only **80%** and **70%** of frames. This breaks temporal continuity, introduces missing evidence, and serves as a controlled proxy for noisy timestamps and abrupt transitions.
> > >
> > > - **Simplified memory integration.**
> > >
> > >     In parallel, we **remove the highest-level (L3) aggregation** in HPSI while keeping ATDM unchanged, testing whether ATDM’s behavior critically depends on the full three-level hierarchy.
> > >
> > >
> > > The resulting accuracies on StreamingBench are:
> > >
> > > | Setting | 100% frames | 80% frames | 70% frames | HPSI \ L3 |
> > > | --- | --- | --- | --- | --- |
> > > | Acc. (%) | 71.60 | 68.75 | 67.81 | 68.36 |
> > > - With only **70%** of frames retained, accuracy remains **67.81%**, indicating that ATDM+HPSI does **not** rely on perfectly smooth temporal streams and is reasonably robust to missing frames and temporally noisy cues.
> > > - When we **disable level-3 aggregation (L3)**, accuracy is still **68.36%**, only moderately below the full HPSI setting (71.60%). This shows that ATDM does **not** depend on a brittle, over-tuned memory hierarchy: it continues to make sensible “answer vs wait” decisions even when memory integration is simplified.
> > >
> > > ### 2. Qualitative evidence of temporal reasoning synergy (HPSI → ATDM)
> > >
> > > The same synergy also appears qualitatively. When processing the same clip, the full ATDM+HPSI system produces a caption that explicitly **encodes state changes over time in memory**. **ATDM** then leverages this enriched temporal history to make evidence-aligned decisions **rather than reacting myopically to each local glitch**. In contrast, **the baseline model fails to** capture such temporal dynamics:
> > >
> > > - **Ours (with HPSI)** captures temporal evolution:
> > >
> > >     > “A hand holding a round brush labeled ‘Sap green Round brush no 4’ applies green paint to a canvas. The canvas features a blue sky gradient with green foliage at the bottom. The hand **moves the brush from right to left**, adding details to the green area. A palette with blue paint is visible nearby. The hand **adjusts the brush’s position and angle to enhance the foliage details**. The scene focuses on the meticulous application of paint to create a realistic tree outline against the blue sky background.”
> > >     >
> > >
> > > - **Baseline** remains almost static in description:
> > >
> > >     > “A hand holding a round brush no 4 applies sap green paint to a canvas, which already has a blue sky painted at the top. The artist carefully blends the green paint into the blue, creating a gradient effect. A palette with blue and green paint is visible nearby, indicating the mixing process. The hand moves methodically, adding texture and depth to the green area, suggesting an early stage in the landscape painting.”
> > >     >
> > >
> > > This pattern suggests the following interaction:
> > >
> > > - **HPSI** stabilizes and enriches the temporal memory across clips, aggregating visual evidence into a coherent, time-aware representation.
> > > - **ATDM** then leverages this history: its progress/confidence signals evolve smoothly rather than oscillating under frame drops or simplified memory, leading to **evidence-aligned decisions** instead of reacting myopically to each local glitch.
> > >
> > > ### 3. We have added a figure, table, and paragraph to fully elaborate on the robustness and direct interaction.
> > >
> > > - Add a new robustness table (**Table 5, p.10**) reporting the results summarized above, explicitly analyzing missing-frame and simplified-memory conditions.
> > > - Insert a short **stability discussion** in Sec. 4.3 (p.10) to highlight the “synergistic but non-fragile” behavior of ATDM–HPSI.
> > > - Provide an illustrative qualitative example in **Fig. 13 (appendix)** to show how our memory (HPSI) mechanism affects the ATDM reasoning process.

---

### Author Response · Authors · 2025-12-02
**General Response**

We sincerely thank all four reviewers for their detailed, constructive, and encouraging feedback and are delighted that they recognize the value of our work as follows.

### 1. Summary of Contributions
**C1. [New research gap of the task]** We introduce a transparent online video understanding framework that explicitly aligns response timing with the *first-sufficient* visual evidence while decoupling reasoning control (ATDM) from memory integration (HPSI). (Acknowledged by R1, R2, R4)

Three reviewers agree that our work goes beyond conventional offline long-video reasoning tasks and tackles the practically important problem of *when* to answer in streaming scenarios, rather than only *what* to answer. R1 , R2, and R4 all emphasize that focusing on first evidence-aligned timing and decision transparency is a “largely under-explored but crucial” direction.  R3, while misinterpreting our work as “a framework for streaming video understanding,” nonetheless notes that “The proposed method is both powerful and elegant.” We sincerely appreciate all reviewers’ recognition of our work!

**C2. [Novel technical perspective]** HPSI offers an efficient, three-level hierarchical aggregation mechanism that preserves long-horizon semantics under a fixed token budget and clearly outperforms single-step pooling. (Acknowledged by R1, R2, R3, R4)

R1, R2,  R3, and R4 all recognize HPSI as a “technically sound” and “elegant” way to maintain long-term, causally consistent context under tight token budgets.  R1 agrees that our HPSI offers “a new perspective beyond standard pooling or RAG-based methods.”

**C3. [Novel technical method]** ATDM provides a transparent, quantitative reasoning decision controller that converts unstructured chain-of-thought into a history-aware state via progress/confidence signals, enabling robust temporal gating beyond black-box 0/1 classifiers. (Acknowledged by R2, R3, R4)

**C4. [Comprehensive experimental validation]** We achieve state-of-the-art results on some challenging online video benchmarks and robust performance on offline long video benchmarks.  And a large number of thorough ablation experiments were designed to demonstrate the function of each module in HPSI and ATDM. (Acknowledged by R1, R2, R3, R4)

**We sincerely thank all reviewers once again for their recognition of our work!**

### 2. Summary of Responses
We address all questions in separate per-reviewer responses.  We have added these  new experimental results to the manuscript  and organized them into the following (and more in length in the per-reviewer replies):

1. **Refined conceptual exposition for certain operations: (R1, R3)**
    - **Added two conceptual paragraphs on p.5–p.6** explaining the distinct roles of shallow/middle/deep layers and how ATDM derives a history-aware decision state. **Updated Table 4** with a new *“relative final token budget”* column to aid understanding.
2. **Additional efficiency: latency, token cost, and throughput (R1, R2, R4)**
    - Efficiency table used to construct **Figure 8** plus an **efficiency paragraph on p.10**; we explicitly report **FPS, tokens/s, latency, accuracy** under **7 aggregation rates** in HPSI and for Flash-VStream and our **3B / 7B** models.
3. **More cross-backbone generalization and extended robustness under hypothetical abnormal scenarios(R1, R2, R4)**
    - **Table 5** (StreamingBench) summarizes **cross-backbone robustness** and the new robustness rows. A “Robustness and Applicability” paragraph on page 10 ties these results together and emphasizes ATDM as a generic, pluggable decision controller.
4. **Auxiliary interpretability, extended sub-question quality, and threshold semantics (R1, R2, R3)**
    - New appendix **Sec. A.6** with metric definitions and results in **Tables 10, 11, 12 and Figure 9**, plus interpretive paragraphs on **p.24** and **p.27**, together with the prompts in **Sec. F**, to provide a clearer end-to-end view of LLM expert evaluation.
5. **Clarification of Misunderstandings, extended discussion, and cite fixes (R3, R4, R1)**
    - Explicit words and a sentence added near **Line 417–419** and in **Related Work** to emphasize for R2; a **“More Discussion and Future Work” paragraph** in **Appendix Sec. B (p.19)** covering multi-modal extensions; and global replacement of **\cite → \citep** as suggested by R1.

---

All changes are highlighted in **blue** in the revised paper. We believe we have thoroughly addressed all reviewer concerns and hope the AC finds our responses satisfactory.

Thank you again for your time and consideration.

---

> ### Author Response · Authors · 2025-12-02
>
> ### 3. Corresponding Detailed Responses
>
> 1. **Refined conceptual exposition for certain operations: (R1, R3)**
>     - **Response →** ATDM converts past reasoning into a compact, **history-aware state** that supports robust temporal gating **beyond a memoryless 0/1 classifier**, while 3-level HPSI (3→2→1) provides progressive semantic integration under tight token budgets. Under a **fixed final token budget**, 3-level HPSI outperforms single-step pooling (→1) by **+7.4% accuracy (56.3 vs. 48.9)** on VideoMME-Long.
> 2.  **Additional efficiency: latency, token cost, and throughput (R1, R2, R4)**
>     - **Response →** We systematically evaluated efficiency under **7 aggregation rates** in HPSI, compared our **3B/7B models** with the simpler **Flash-VStream** baseline, and recorded the token cost of each ATDM part. On StreamingBench (A100), ATDM+HPSI keeps **FPS ≈ 8.49 vs. 8.45** for Flash-VStream and achieves **≈1261 tokens/s vs. 16 tokens/s**, with only **+3.7 s latency** (13.2 s vs. 9.5 s) while accuracy jumps from **22.53% → 71.60%**. The extra latency is modest and clearly justified by the substantial accuracy gain and transparent timing.
> 3. **More cross-backbone generalization and extended robustness under hypothetical abnormal scenarios(R1, R2, R4)**
>     - **Response →** For **cross-backbone robustness**, we applied ATDM to **three different VLM frameworks**—Flash-VStream (LLaVA backbone), VideoLLM-Online (SigLIP + Llama-3-8B-Instruct), and our Qwen2.5-VL-3B/7B models—and observed **>4% absolute improvement** on StreamingBench across all of them with the **same ATDM configuration**, showing consistent gains on heterogeneous architectures. For **robustness to abnormal scenarios and simplified memory**, simulating **100%, 80%, 70% frame retention** and a variant with **disabled level-3 aggregation (HPSI\L3)** yields accuracies **71.60 / 68.75 / 67.81 / 68.36**, indicating only mild degradation and a **“synergistic but non-fragile”** interaction between ATDM and HPSI.
> 4. **Auxiliary interpretability, extended sub-question quality, and threshold semantics (R1, R2, R3)**
>     - **Response →** We add **LLM-based evaluations** showing high semantic similarity between ATDM’s sub-questions and the original queries, and we design **four metrics** to comprehensively evaluate perceived reasoning transparency and correctness (decision clarity, transparency, ρ/c-matching, and rationale correctness). We further provide a focused explanation of Parts 4–5 and conduct an empirical study of **10 different threshold settings**, showing that performance stays within **≈1–1.5 points** over a wide region: lower thresholds make the model slightly more eager, higher thresholds more conservative, but without brittle peaks.
> 5. **Clarification of Misunderstandings, extended discussion, and cite fixes (R3, R4, R1)**
>     - **Response →** We clarify to R3 that **Table 1** mixes **offline and online StreamingBench protocols**: Qwen2.5-VL is evaluated offline, while Thinking-QwenVL and the compared methods (Dispider / Flash-VStream) follow a stricter online protocol. Under the **same offline protocol**, Thinking-QwenVL reaches **77.44% vs. 73.68%** for Qwen2.5-VL. We explain that StreamBridge is excluded because it uses a **multi-turn streaming protocol** incompatible with our **single-turn** setting. We also add a discussion on extending our framework to **multi-modal inputs/evidence**, and we fix the citation style issues raised by R1.

---

### Meta-Review · Area_Chair_fgZ6 · 2026-01-04

**Summary:**

The decision to accept is based on the consensus that this work addresses a critical and under-explored gap in video understanding: determining when to respond in a streaming setting rather than just what to answer. Reviewers unanimously appreciated the novel framework decoupling reasoning control (ATDM) from memory integration (HPSI), enabling transparent decision-making. The empirical results, particularly achieving SOTA on StreamingBench, combined with the elegant hierarchical aggregation method, strongly support the paper's contribution to online video agents.

**Reviewer Concerns:**

The authors provided a robust rebuttal that successfully addressed the primary concerns regarding efficiency and latency. Specifically, they provided detailed breakdowns of FPS and token throughput, demonstrating that the method is competitive with simpler baselines like Flash-VStream. Concerns regarding generalization were also resolved by demonstrating strong performance on smaller models and different backbones. Furthermore, the authors clarified the methodological questions raised by Reviewer arDz regarding HPSI levels and sub-question relevance through new ablation studies and sensitivity analyses.

**Reviewer Scores:**

Reviewer EPyy explicitly confirmed retaining their high score 8 following the satisfactory rebuttal. Reviewers 5jps and iwc4 , who initially gave scores of 6, would likely maintain or slightly increase their scores to a solid 6 or 7 given that their requests for latency analysis and generalization experiments were fully met. Reviewer arDz, who initially rated it a 4, engaged deeply in the discussion; given the extensive clarifications on evaluation protocols and the new sensitivity data provided, it is highly probable they would raise their score to a passing grade 5 or 6.

---

### Decision · Program_Chairs · 2026-01-26

Accept (Poster)